# Maternal transmission as a microbial symbiont sieve, and the absence of lactation in male mammals

Brennen T. Fagan [1,2] ✉, George W. A. Constable [2] & Richard Law [2]

Gut microbiomes of mammals carry a complex symbiotic assemblage of microorganisms. Feeding newborn infants milk from the mammary gland allows vertical transmission of the parental milk microbiome to the offspring's gut microbiome. This has benefits, but also has hazards for the host population. Using mathematical models, we demonstrate that biparental vertical transmission enables deleterious microbial elements to invade host populations. In contrast, uniparental vertical transmission acts as a sieve, preventing these invasions. Moreover, we show that deleterious symbionts generate selection on host modifier genes that keep uniparental transmission in place. Since microbial transmission occurs during birth in placental mammals, subsequent transmission of the milk microbiome needs to be maternal to avoid the spread of deleterious elements. This paper therefore argues that viviparity and the hazards from biparental transmission of the milk microbiome, together generate selection against male lactation in placental mammals.

The absence of male lactation in mammals is a puzzle—there appears to be no universally convincing reason why it should not happen. John Maynard Smith pointed out that paternal care that incorporates such feeding would be evolutionarily stable in monogamous mammals[1]. There have been over 200 My for male lactation to evolve[2]. Genetic control of the mammary gland is widely distributed across mammalian chromosomes[3]. Genetically male mammals have mammary tissue. They are known to have the potential to lactate[4–8], and milk production has been recorded under natural conditions in the Dayak fruit bat (*Dyacopterus spadiceus*)[4]. Lactation requires hormonal triggers to take place, such as high levels of prolactin, which are generally down-regulated in males, preventing this from happening [e.g. ref. 9] It seems there are selection pressures preventing male lactation.

A well-known answer to the puzzle, building on the work of Trivers[10,11], is that the absence of male lactation is simply a result of selection for sex-biased parental care. Should paternity be uncertain[12,13], competition for female mates comparatively low[14], or sexual selection on males high[15], an evolutionary pressure exists for males to abandon care for their young in favour of additional mating opportunities. In such situations (in which male parental care is

selected against), it is clear that the evolution of male lactation should be likewise prevented. However, should a combination of the above conditions not hold (i.e. paternity is more certain, competition for mates high, or sexual selection low), biparental care can instead evolve[15]. Indeed, this is the situation in around 10% of mammals[16], amongst which Azara's owl monkeys (*Aotus azarae*) are perhaps the starkest example; here the male is almost certain to be the father and the male is responsible for almost all care except for nursing[17,18]. This leads to the question of why, given other forms of paternal care have evolved[19,20], male lactation remains the rare exception rather than the rule in these socially monogamous mammals.

This paper draws attention to a further set of participants in mammalian lactation which have important effects on the role of males. These are the microorganisms that form the milk microbiome, and that are transmitted from parent to offspring during feeding[21]. We call the association 'symbiotic' because this term is widely used to describe intimate and prolonged physical associations between dissimilar organisms—here a mammalian host and a microbial symbiont—irrespective of where the association lies on the mutualism-parasitism continuum[22,23]. We note that the term symbiotic is sometimes used as

[1]Leverhulme Centre for Anthropocene Biodiversity, University of York, York, UK. [2]Department of Mathematics, University of York, York, UK.
✉e-mail: Brennen.Fagan@york.ac.uk

shorthand for mutualistic symbiotic interactions in the microbial literature [ref. 24, but see ref. 25], but we need the more general usage here, because this paper is concerned with how variation in the interactions drives natural selection on the host's vertical transmission of microbes. These microbial symbionts could provide an evolutionary explanation for the absence of lactation in male mammals even when males invest in other forms of parental care.

Vertical transmission of symbionts through lactation, during the period from birth to weaning, provides a reliable conduit for moving microbes from host parent into the gut of host offspring. This is in contrast to horizontal transmission where symbionts are taken up from the environment in a less targeted way (see[25] for a review of symbiont transmission). We focus on mammalian milk here, which contains its own microbiome, including bacteria, fungi and viruses[21]. Although just one of a number of channels for transmitting microbes from mammalian parents into the gut of their infants, milk is thought to make a major contribution to the infant's gut microbiome early in life[26], with ~$10^3$–$10^4$ colony-forming units of bacteria ml$^{-1}$ in the case of human milk[27]. Vertical transmission of elements of the microbiome is well documented[28]; it is known to work across multiple host generations in mice[29], and down to the level of specific clusters of parental strains of bacteria in the case of human milk[30,31].

However, there is a basic, general danger to the host population from biparental transmission of symbionts. This is seen in its strongest form when a rare symbiont first colonises a host population, i.e. when most matings by hosts carrying the symbiont are with uninfected hosts (Fig. 1). When transmission is biparental, the symbiont is transmitted if it is carried by either host parent. In contrast, when transmission is uniparental (usually maternal), the symbiont has only one transmission route. Biparental transmission gives the symbiont a twofold reproductive boost when it is rare, enabling it to invade a host population, even if it is harmful to the host. (We note that harmful is a relative measure here, comparing the fitness of the host that carries the symbiont to the fitness of a host that does not.) Uniparental transmission removes this boost. This was recognised in early work on the evolution of uniparental cytoplasmic inheritance[32,33], and was extended to the mixing of symbiotic lineages[34], but not to vertical transmission of the gut microbiome. We develop our argument in the context of male lactation in mammals, but note that it may apply much more widely because mother-to-infant is thought to be the usual channel for symbiont transmission in animals with sexual reproduction[35].

Here we use mathematical models to show that maternal transmission operates as a sieve on vertically transmitted symbionts. This sieve prevents invasion by a class of symbionts with deleterious effects on their hosts that would otherwise spread under biparental

transmission. At the same time, the sieve still permits spread of symbionts with beneficial effects on their hosts. We demonstrate that deleterious symbionts also generate a selective advantage for host modifier genes preventing vertical transmission of symbionts through male hosts, and show that this selective advantage is maintained in the presence of some horizontal transmission of the deleterious symbionts. We note the significance of these results to the absence of male lactation in mammalian hosts: transmission of the milk microbiome to the gut microbiome of offspring is almost invariably from the mother. Although biparental milk production could bring nutritional benefits to offspring, we show that these benefits can be outweighed by the costs associated with deleterious symbionts.

## Results
### The symbiont sieve
Consider a host's symbiont community comprising a set of microbial taxa, labelled $s$. Suppose a new microbial taxon is added to the resident set of symbionts, creating a new community, labelled $S$. The new symbiont is initially rare, so the community $S$ starts at a low frequency in the host population. We examine how the fate of $S$ is determined by host's mode of vertical transmission, focusing on the fate of the new symbiont by making two simplifying assumptions: (1) it is independent of the resident microbiome, and (2) there is no evolution in the microbiome. Interactions within the microbiome are of course important[36], as is its evolution[37,38], but these are not needed to demonstrate the simple contrasting effects of uniparental and biparental transmission. (See Methods: Relation to other modelling frameworks; microbes with no vertical transmission lie outside the scope of this study.)

The advantage to the hosts of restricting symbiont transmission to one rather than both parents can be made precise with a little algebra[33]. Denote $w$ as the fitness of hosts carrying the new symbiont (community $S$) relative to the fitness of those not carrying it (community $s$). This means the new symbiont is beneficial to a host if $w > 1$, and deleterious if $w < 1$. Under maternal transmission, the symbiont is transmitted only if the mother is carrying it. However under biparental transmission, it is transmitted if either or both parents are carrying it. We show that the condition for the invasion of the symbiont is

$$
\begin{aligned}
w &> 1 && \text{(maternal transmission)}\\
w &> \tfrac{1}{2} && \text{(biparental transmission)}
\end{aligned}
\tag{1}
$$

(see Methods: Algebraic model). So maternal transmission restricts invasion of host populations by symbionts to those that are beneficial. In contrast, biparental transmission allows invasion by symbionts that are deleterious, so long as the relative fitness of hosts remains greater than one half. This demonstrates the reproductive boost that allows symbionts to invade under biparental transmission, even if they reduce host fitness.

A similar argument can be constructed to show that transmission of symbionts through the father rather than the mother would be just as advantageous. However, viviparity in placental mammals makes some transmission of microorganisms from the mother to infant unavoidable. Evidence for this includes differences observed in the gut microbiomes of human infants born vaginally, and those delivered by caesarean section[39], which implies some transfer of symbionts occurs during a vaginal birth[35]. There are also indications that bacteria can reach the uterus through the blood stream of the mother in mice[40] (although the paradigm of the sterile womb in humans does still have support[41,42]). If transmission is to be restricted to one parent in placental mammals it needs to be the mother.

Figure 2 illustrates the dramatic effect of what could be called the symbiont sieve that comes from maternal transmission. It shows the ultimate fate (fixation or loss) of a new symbiont (community $S$) giving host fitness $w$ relative to hosts without it ($s$), under biparental and

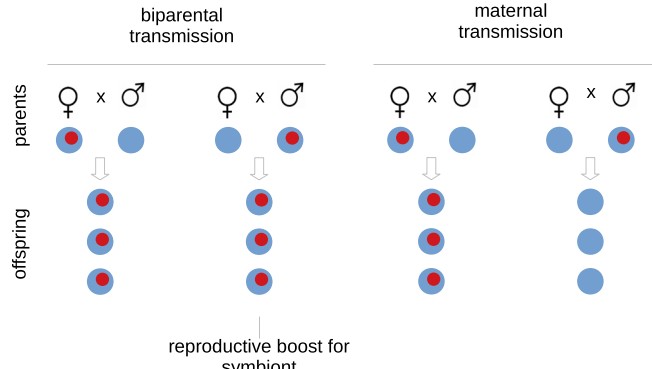

| biparental transmission | maternal transmission |

Fig. 1 | **Biparental transmission of symbionts in a host population gives the symbionts a reproductive boost.** This is at its greatest when symbionts (in red) are rare in the host population, and infected hosts mate mostly with uninfected ones, as shown here. The boost is prevented by uniparental transmission, assumed here to be maternal.

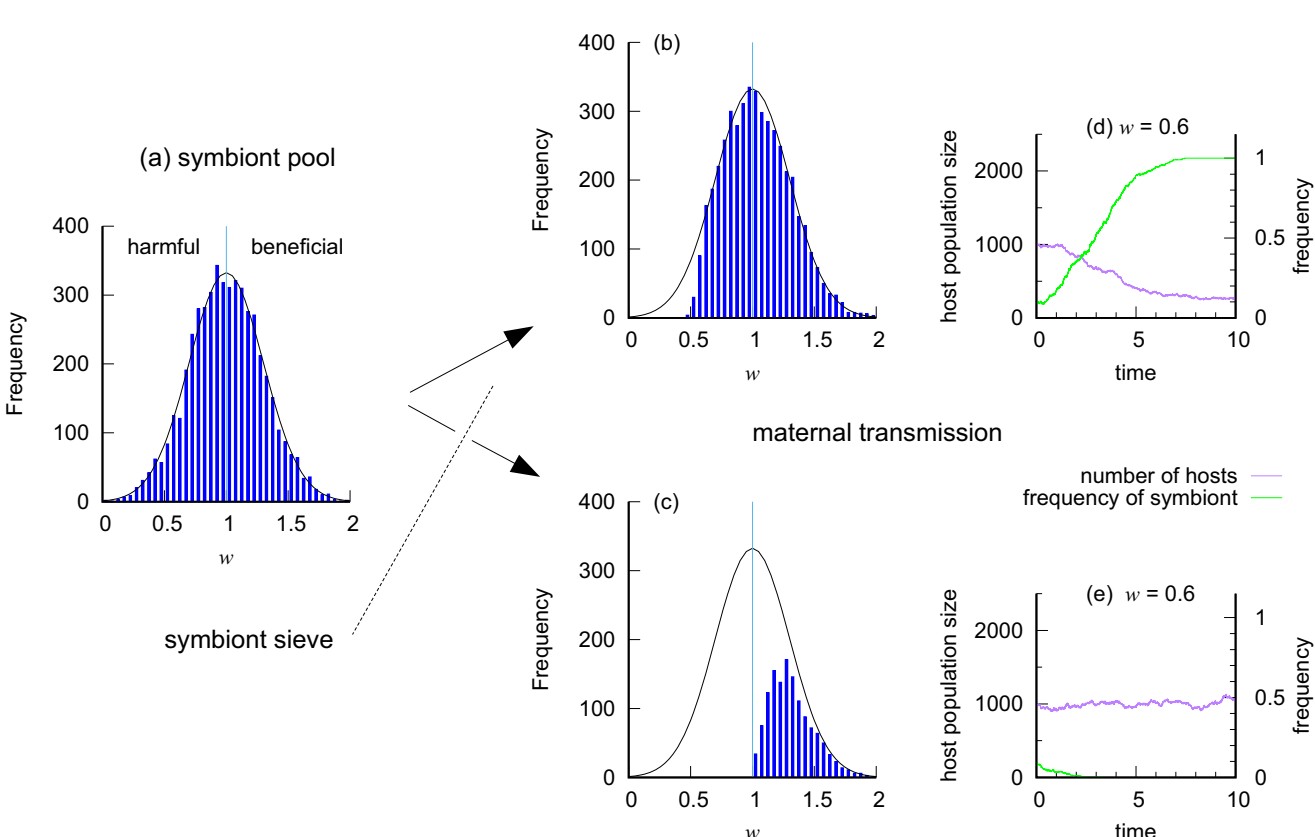

**Fig. 2 | A numerical experiment to demonstrate host maternal transmission operating as a symbiont sieve. a** A (truncated) normal frequency distribution from which to draw random values of $w$, the effect of a new symbiont (community $S$) on host fitness. The symbiont is harmful if $w < 1$, and beneficial if $w > 1$. **b** Biparental transmission allows some harmful symbionts to establish in the host population. **c** Maternal transmission sieves out all harmful symbionts, only allowing beneficial symbionts to establish. Distributions (**b**) and (**c**) were created from 5000 independent draws of $w$ from (**a**). **d, e** Two example time series with $w = 0.6$. Source data are provided as Source Data files[77].

maternal transmission. We embedded the algebraic model into a stochastic birth-death process of the host population to do this (see Methods: Stochastic birth-death model). This is more nuanced than the algebraic model, having a wider range of outcomes including extinction of the host population driven by deleterious symbionts.

Figure 2a describes an initial distribution of fitness $w$ in a pool from which to draw symbionts. Figure 2b shows the distribution of symbionts that successfully invade host populations during a birth-death process with biparental transmission; deleterious symbionts can sweep through the host population, as long as they no more than double the death rate of hosts (i.e. $w > 1/2$), as in the simpler algebraic model. In contrast, maternal transmission (Fig. 2c) operates as a sieve, preventing invasion by deleterious symbionts ($w < 1$), while allowing the beneficial ones ($w > 1$) to spread to fixation. Examples of the time series with $w$ set to 0.6 (Fig. 2d, e) show the symbiont (community $S$) increasing in frequency under biparental transmission and decreasing under maternal transmission. Population size falls as the symbiont increases because it causes an increase in host death rate (see Methods: Stochastic birth-death model). Near the point of neutrality under maternal transmission ($w = 1$), demographic stochasticity often leads to extinction of the symbiont before it can establish. In an infinitely large population, the probability of invasion would tend to zero as $w \to 1$ from above.

A similar principle operates when multiple new symbiont taxa are simultaneously introduced (Fig. 3). The example shown here describes the fate of twenty new symbiont taxa, some beneficial and others

deleterious to their hosts. On one hand, the reproductive boost from biparental transmission allows rapid spread of all the symbiont taxa, beneficial and deleterious, through all hosts (Fig. 3a). This is because transmission through both parents allows the taxa to mix freely across hosts. On the other hand, maternal transmission stops this mixing from taking place (Fig. 3e). The frequency distributions (Fig. 3b–d and f–h) show the number of symbiont taxa present in the hosts. At time 0 the two modes of transmission are initialised to the same distribution of microbiomes (panels b, f). But they diverge rapidly (panels c, g). By the end of the time period shown, almost all hosts carry every symbiont when transmission is biparental (panel d). In contrast, almost all hosts carry the single taxon which confers the greatest benefit to the host when transmission is maternal (panel h).

Note that the protection given by maternal transmission has the cost that beneficial symbionts spread more slowly than they would under biparental transmission (note the different time scales in Fig. 3a and e). How significant this is depends on the distribution of host fitnesses generated by the pool of available symbionts. The distribution is unknown, but it most likely skews towards symbionts with deleterious effects ($w < 1$), after a sequence of successful invasions has taken place. By this stage, protection from deleterious symbionts is likely to be more important.

## Evolution of the symbiont sieve
How symbionts drive the evolution of vertical transmission by their hosts is a separate and important matter. Here we show that

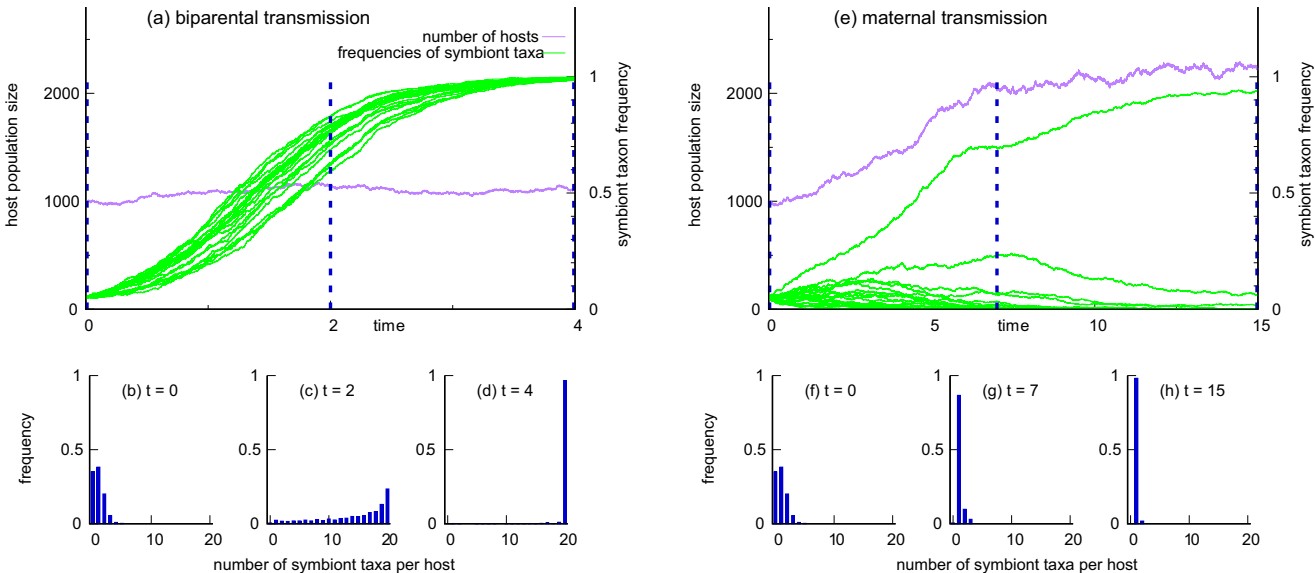

**Fig. 3 | Effect of host transmission mode on invasion by 20 symbiont taxa.**
**a** Biparental transmission in this instance allows all symbiont taxa to become established, whether harmful or beneficial. **b**–**d** The host population moves towards a state in which every host carries every symbiont taxon. **e** Maternal transmission prevents mixing of symbiont taxa, and no new combinations of taxa are created after inoculation at time 0. **f**–**h** Hosts carrying the single most beneficial symbiont replace all others, and the host population moves towards a state in which every host carries just the most beneficial symbiont taxon; this is accompanied by an increase in host population size. The 20 symbiont taxa were generated by drawing independent values of $w_i$, the effect of taxon $i$ on host fitness, from the distribution in Fig. 2a. The overall effect on a host was taken as the average value of $w_i$ over the symbiont taxa present in the host. The simulations were started by randomly inoculating 50 of 1000 hosts with one symbiont taxon, repeating this independently for each of the 20 taxa. Source data are provided as Source Data files[77].

uniparental transmission is adaptive in the presence of deleterious symbionts. The symbionts can cause vertical transmission in their hosts to evolve from biparental to uniparental transmission and prevent the transition back to biparental transmission once it is uniparental. The symbionts do this through selection on host modifier genes that affect the transmission of the host's symbionts, analogous to modifier genes that influence the inheritance of genes. Selection on such modifiers operates while a deleterious symbiont is spreading under biparental transmission (Fig. 3a), i.e. when the symbiont is present in some hosts and absent in others. It is intuitive that, if the father carries a deleterious symbiont and the mother does not, a host gene that stops transmission from the father will gain an advantage. Selection on such host genes then drives evolution of vertical transmission in the host population.

To illustrate evolution of maternal transmission, Fig. 4 uses the stochastic birth-death model (see Methods: Stochastic birth-death model), with two alternative modifier genes $\{M^+, M^-\}$ to control vertical transmission by male hosts, during the invasion of a host population by a deleterious symbiont ($w = 0.6$). (Transmission through females is always present.) For simplicity, we assumed the modifier genes are at a locus on the Y chromosome, $M^+$ allowing transmission from males making vertical transmission biparental, and $M^-$ preventing transmission from males making vertical transmission maternal.

Starting with a low frequency of $M^-$, a deleterious symbiont can invade the host population, because vertical transmission is predominantly biparental (Fig. 4a). $M^-$ then increases in frequency as it becomes associated with hosts lacking the symbiont, which have greater fitness than hosts carrying the symbiont. This association can be measured by a coefficient of disequilibrium $D$ (see Methods: Stochastic birth-death model), which becomes positive (Fig. 4b). The association drives the host population towards maternal transmission, and eventually $M^-$ reaches a frequency great enough to turn the tables against the deleterious symbiont. $M^-$ goes to fixation, making vertical transmission fully maternal, and the deleterious symbiont is eliminated from the host population. The outcome is a host population with both maternal transmission and protection from harmful symbiont with $1 > w > 1/2$.

An alternative scenario is that $M^-$ does not reach fixation before the symbiont is present or absent in all hosts (see Fig 1 of ref. 33). In this case, the path to maternal transmission goes in steps: first $M^-$ increases whenever a deleterious symbiont is present in some but not all hosts; then on fixation (or loss) of the deleterious symbiont, $M^-$ and $M^+$ are neutral (hosts all carry the same set of symbionts, so the mode of transmission has no effect on host fitness). Selection for $M^-$ is triggered again by the arrival of a new deleterious symbiont. Intermittent selection on $M^-$ continues in this way until $M^-$ is fixed and maternal transmission occurs throughout the host population.

Evolution from biparental to maternal transmission sheds light on host-symbiont evolution. It is not vertical transmission itself that checks the spread of deleterious symbionts in host populations with separate sexes. Rather it is the restriction of vertical transmission to just one sex that makes the symbiont sieve work.

## Horizontal transmission and the symbiont sieve

It needs to be kept in mind that host populations are still vulnerable to invasion by a deleterious symbiont if the symbiont can be acquired by horizontal transmission from the environment, as well as by vertical transmission. We examine this in Fig. 5, extending our model to allow a fixed per-capita rate $e_0 > 0$ at which hosts acquire the new symbiont from an environmental source[43]. An environmental input into the host's microbial community is potentially important, and means that the community is no longer fixed for the lifetime of the host (we do not deal with direct horizontal transmission from host to host). Figure 5 makes use of a system of differential equations describing the average behaviour of the stochastic birth-death model (see Methods: Differential equation model). As would be expected, it shows that horizontal transmission will overwhelm vertical transmission if $e_0$ is sufficiently large. However, if $e_0$ is of the same order as birth and death rates, vertical transmission exerts some control over the symbiont community, and the outcome then depends on whether vertical transmission is biparental or maternal (uniparental).

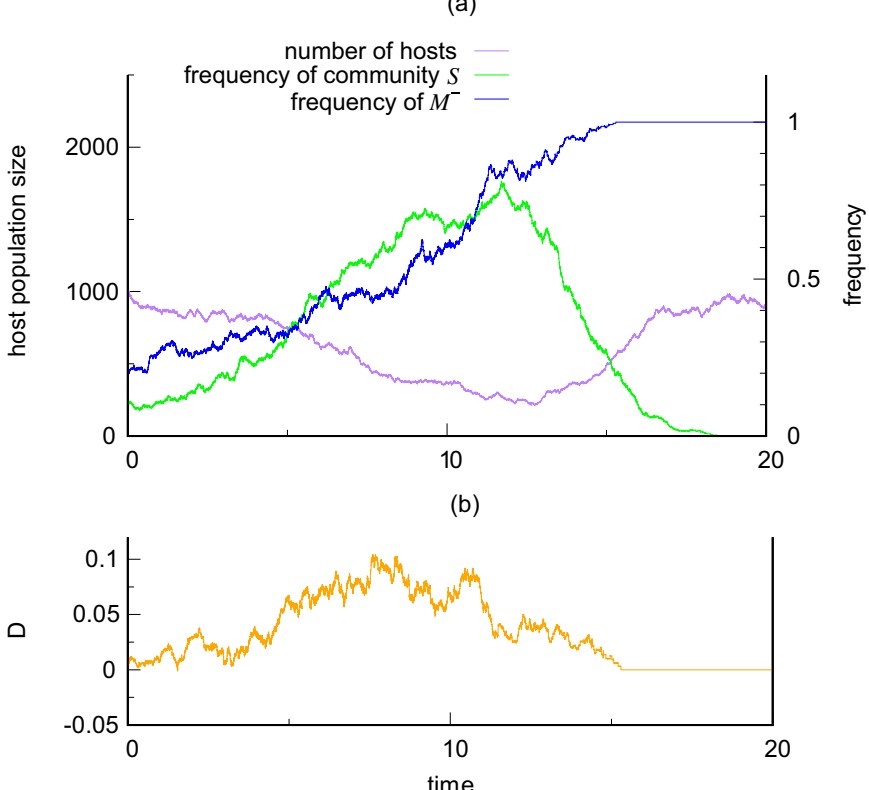

**Fig. 4 | A harmful symbiont ($w = 0.6$) drives evolution of maternal transmission.** **a** Here a host population starts mostly with biparental transmission (high frequency of host gene $M^+$, low frequency of $M^-$), and a low frequency of the symbiont community $S$ in the host population. This allows the new deleterious symbiont to spread. The symbiont generates a selective advantage for the $M^-$ gene that prevents male transmission. The host population then becomes dominated by maternal transmission $M^-$, and under these conditions the symbiont cannot persist. **b** The gene $M^-$ gains its advantage by becoming associated with male hosts that lack the symbiont, as measured by the coefficient of disequilibrium $D$. Source data are provided as a Source Data file[77].

Specifically Fig. 5a, b shows a region now exists that prevents a deleterious symbiont from going to fixation. Instead, when $e_0 > 0$ the host population goes to a bimorphic equilibrium at which there are two host types present: those that carry the new harmful symbiont, and those that do not. This is important because the host genes modifying vertical transmission need both types of host to be present for natural selection to operate on them. Horizontal and biparental transmission together (Fig. 5a) then result in continuing selection for the modifier gene $M^-$ that suppresses transmission through males. The host population then evolves to a state where all vertical transmission is maternal in the presence of horizontal transmission (Fig. 5b), as illustrated by the path in Fig. 5c. There is no return route to biparental transmission because the modifier gene $M^-$ benefits from the bimorphic state. Horizontal transmission in effect helps to protect maternal transmission. (Figure 5c also gives the time solution from the differential equations corresponding to the stochastic process, to illustrate that they capture correctly the general features of the stochastic process.)

Note that there is a region in Fig. 5a in which the reproductive boost to the symbiont from biparental transmission, alongside its harmful effect on the host, drives the host population to extinction. This region does not exist under maternal transmission for $e_0 < 1$, and emphasizes the extra dangers resulting from biparental transmission in the presence of deleterious symbionts.

In the third region of Fig. 5a and b, the new symbiont goes to fixation, and the subset $w < 1$ of this region is the range over which the successful symbiont is harmful to the host. This subset is independent of $e_0$ under biparental transmission, and includes $e_0 = 0$. However, this region gets smaller as $e_0$ decreases under maternal transmission, and

the symbiont sieve fully protects the host population when transmission is strictly vertical (i.e. $e_0 = 0$). Horizontal transmission dominates when $e_0 \geq 1$, irrespective of whether vertical transmission is biparental or maternal, suggesting that the constant churn created by horizontal transmission renders biparental and maternal transmission neutral with respect to the microbiome (but not to other ways of increasing fitness, such as care). Also, beneficial symbionts ($w > 1$) go to fixation irrespective of the type of vertical transmission, and they do so faster when this transmission is biparental.

To summarise, suppose a symbiont is transmitted both horizontally and biparentally. A strongly deleterious symbiont ($w < 0.5$), either leads to collapse of the host population and is therefore rarely seen, or it leads to a weakened bimorphic host population if the rate of horizontal transmission is low enough giving maternal transmission a continual advantage. A weakly deleterious symbiont ($0.5 < w < 1$) always spreads to fixation, but still leaves a transient period during which the host population is bimorphic allowing selection for maternal transmission. In other words, even with the addition of horizontal transmission, deleterious symbionts typically turn the ratchet towards more maternal transmission and less transmission through males.

## Biparental nutrition and the symbiont sieve
It is important to recognise that provision of additional milk to the infant by the father could increase the total supply of food to the infant. Therefore biparental lactation has the potential to be beneficial through the nutritional role of mammary milk. Yet, nursing by males remains a very rare exception in mammals, even when males participate extensively in many other aspects of offspring care. Why then are

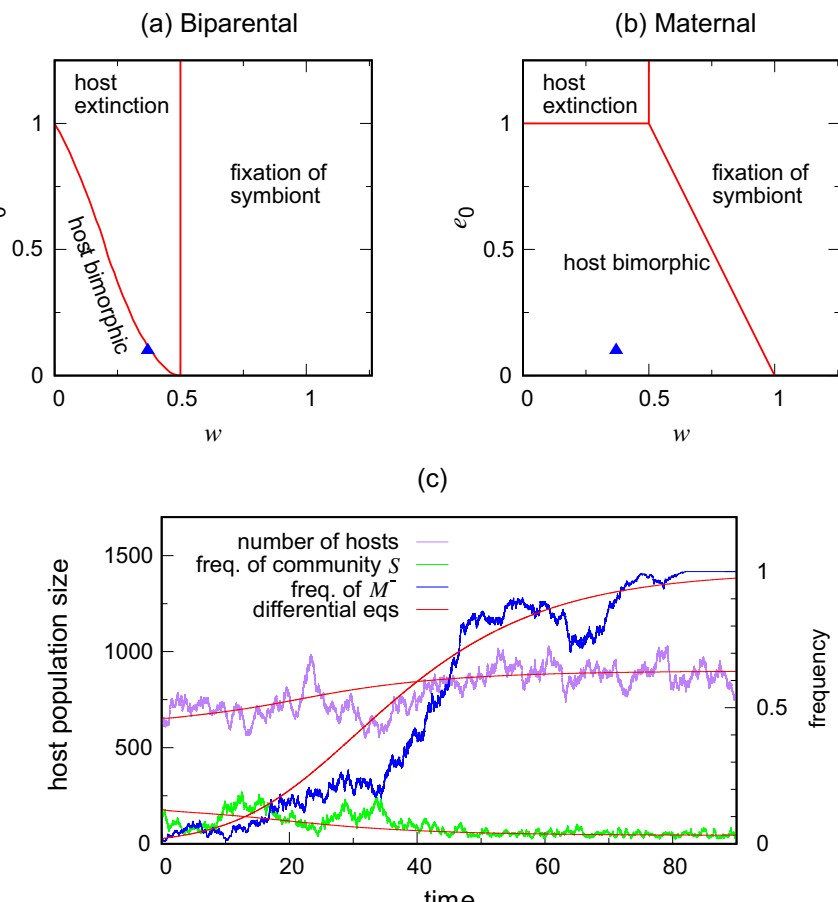

**Fig. 5 | Three contrasting outcomes following arrival of a new symbiont with horizontal transmission under biparental and maternal vertical transmission.** Horizontal transmission takes place at a fixed per-capita rate $e_0$, and $w$ is the effect of the new symbiont on host fitness. **c** Shows that the bimorphic state allows selection for a host gene $M^-$ causing loss of male transmission, changing host vertical transmission from biparental to maternal; $(w, e_0) = (0.37, 0.1)$ here, the triangle in (**a**) and (**b**). Outcomes in (**a**) and (**b**) were computed by numerically solving a pair of coupled differential eqs. (5) and (6), and (**c**) from a stochastic birth-death process overlaid with the corresponding solution from a system of six coupled differential equations (7) to (12). See Methods for the equations. Source data are provided as Source Data files[77].

the potential benefits of biparental feeding insufficient to outweigh the benefits of the symbiont sieve?

To address this requires a further trait: milk volume. We note a basic biological asymmetry between (a) milk as a source of food, and (b) milk as a channel for vertical transmission of microorganisms. On one hand, the nutritional benefits of milk are quantitative, increasing as the milk volume increases, although not necessarily linearly. On the other hand, transmission of symbionts can be achieved even with relatively small milk volumes, because microbial populations can build up after arrival in the infant gut. Since the symbiont sieve is absent when lactation is biparental, these microbes can be deleterious.

The evolution of male lactation then depends on the nutritional benefits of greater milk volume relative to the costs of losing the symbiont sieve. We examine this trade-off in the Supplementary Information (Supplementary Information Section: Modelling the benefits of biparental lactation). The model is based on the effect of extra milk from males on: (a) nutrition which increases infant survival, and (b) the introduction of deleterious symbionts which comes about through loss of the symbiont sieve, reducing infant survival. This extends the model in Section: Evolution of the symbiont sieve. The model now includes a modifier gene $M_\nu^+$ that leads to a milk volume $\nu$ from males and therefore allows infants to feed on and take up symbionts from male mammary milk. As in the earlier model, it also includes a modifier gene $M_{\nu=0}^-$ in which male lactation is absent,

thereby preventing infants from taking up food and symbionts from the father.

Starting from an ancestral state in which lactation is maternal, the model shows that a mutant $M_{\nu=\delta}^+$, that allows the father to provide a small volume $\delta$ of milk to infants, is unable to invade and replace the resident gene $M_{\nu=0}^-$ (Supplementary Fig. 2). This operates under the condition that the benefit to the infant is small, and the cost is relatively large because deleterious symbionts can still become established in the infant's gut (the symbiont sieve is broken). In other words, lactation when confined to mothers is robust to invasion by host mutants that allow a low level of lactation in fathers.

Conversely, starting from an ancestral state in which lactation is fully biparental, with a resident gene $M_{\nu=1}^+$ (i.e. fathers providing as much milk as the mother), the model shows that a mutant $M_{\nu=1-\delta}^+$ that reduces milk production by a small amount $\delta$, is also unable to invade and replace the resident (Supplementary Fig. 3). This operates under the condition that the mutant has little, if any, effect on invasion by deleterious symbionts, whereas the nutritional benefit to the infant is reduced. In other words, fully biparental lactation is also robust to invasion by host mutants with small reductions in male lactation.

These results suggest that uniparental or biparental ancestral states could both be uninvadable, strictly within the context of milk (a) as a source of food, and (b) as a channel for vertical transmission of microorganisms. Other selective pressures would probably be needed to evolve from a biparental ancestral state to one in which lactation is

limited to females. An obvious contender would be the selective advantage to males that abandon care of their young in favour of additional mating opportunities[10,11]. The key point is that, once lactation is confined to females, the symbiont sieve creates a barrier to the evolution of paternal lactation under conditions that favour evolution of biparental care in general. The symbiont sieve therefore provides a possible explanation for the absence of lactation by males, when biparental care evolves from maternal care.

## Discussion

In effect, maternal transmission acts as a first line of defence for hosts against microbes potentially transferable from one host generation to the next. It operates at the start of host life as a symbiont sieve, separating microbes that are beneficial to the host (mutualists) from those that are deleterious (pathogens), preventing deleterious ones from spreading in the host population. Paternal uncertainty must be a major driver for the absence of lactation in male mammals[10,11]. However, the symbiont sieve offers an additional explanation for its continuing absence in mammals which otherwise display well-developed paternal care. The sieve deserves special attention because of its significance for early development of gut microbiomes in infant mammals.

The theory is falsifiable: widespread deleterious symbionts found to be propagating solely through vertical transmission of mammary milk in mammals would be inconsistent with the symbiont sieve. This is not to claim that there are no pathogens transmitted via milk which is clearly incorrect in the case of human breast or chest milk[44], but deleterious symbionts would need additional mechanisms to spread, such as direct horizontal infection from host to host. Viral transmission through breast or chest milk is documented in humans (e.g. human immunodeficiency virus or cytomegalovirus[44]), but there does not appear to be evidence of deleterious viruses for which mammary milk is the sole transmission mode. The existence of such a symbiont would be evidence of a weakness in the symbiont sieve.

The theory obviously requires the existence of a milk microbiome. This matter is settled: metagenomics has shown that complex microbial communities are present in mammary milk[21], including human milk[45,46]. Transmission of the microbiome to the infant is also established in humans[47], the microorganisms becoming a major feature of the infant's gut microbiome in the early stages of its development[39]. The source of the milk microbiome is a subject of current research, with elements possibly arriving in human milk via an entero-mammary pathway[21,47].

The theory requires host modifier genes that affect vertical transmission of their symbionts. The symbionts then drive evolution of vertical transmission in their hosts by generating natural selection on the modifier genes. In practice there are many ways in which this can happen through host physiology, behaviour and social structure. For instance, host genes that regulate hormones promoting lactation, such as prolactin, are potential targets of natural selection. Genes down-regulating such hormones in male hosts would move host populations towards maternal transmission of symbionts, and would be selected. Genes up-regulating such hormones in males would move host populations towards biparental transmission, and would not be selected.

The theory remains robust when nutritional benefits of milk from biparental lactation are taken into account. Our extended model that accounts for milk volume shows that host mutations permitting a low level of male lactation do not invade a host population in which lactation is confined to females, as long as vertical transmission of microbes can take place through relatively small milk volumes. Interestingly, the state of full biparental lactation, in which male and female hosts contribute equally to lactation, is also robust to invasions by mutations that reduce male lactation by a small amount, because such mutations would have little effect on vertical transmission of microbes.

A caveat to this work is that the nutritional benefits of milk to the offspring need eventually to be balanced against the costs of this to the parents. Addressing these costs would require consideration of how other forms of male parental care (such as the carrying of infants) could supplement investment by females in milk production[48]. This has yet to be accounted for in quantitative evolutionary modelling and is a promising avenue for future work.

The theory predicts that symbionts transmitted just by mammary milk from mother to infant should not be deleterious to the infant. This raises a question as to the benefits they provide. Gut microbiomes are open systems colonized within host generations by a diverse set of microorganisms, and are of remarkable complexity[36]. The order in which these systems are assembled leaves a lasting impact on the structure of the microbiome in mice[49], suggesting a key role for those present near the time of birth. In humans, *Bifidobacterium*, which occurs in breast and chest milk, is known to inhibit the growth of pathogenic bacteria and to aid in the digestion of the milk[50], and there is strong evidence for the vertical transfer of these bacteria during breast and chest feeding[51,52]. Early arrival of symbionts through maternal transmission may help in controlling the assembly of gut microbiomes, as microorganisms are later picked up from a wide range of environmental sources.

On a longer evolutionary time scale, mammalian gut microbiomes are notable for the way in which their structure mirrors the phylogenetic relationships of their mammalian hosts (phylosymbiosis). This is thought to be related to special traits of mammals including viviparous birth, milk production and parental care, the milk allowing efficient vertical transmission of microorganisms from the mother to the infant gut, together with immune factors which act as a filter on the set of symbionts[28]. Our results add the feature that the interaction between microbiome and host actively holds in place maternal transmission of the milk microbiome. In doing so, it puts in place a symbiont sieve which helps to select the symbionts beneficial to the host. Slower turnover of the taxonomic composition of the microbiome is a likely outcome, contributing to the match between microbiome structure and host phylogeny.

There are various open questions that stem from this theory. For instance, there is a famous exception to the absence of male milk production, the Dayak fruit bat[4], in which males with functional mammary glands have been observed under natural conditions. Gut microbiomes of bats are unlike those of other mammals. It is notable that their microbiomes have more resemblance to those of birds that fly, showing little evidence of a correlation with host diet or phylogeny[53]. Adaptations for flight could underly this, including reduced gut size, less microbial biomass, more paracellular absorption of nutrients compensating for a small gut, and less anaerobic conditions in the gut allowing more uptake of microbes from the environment[53]. If the gut microbiome is relatively unimportant, this might reduce the strength of selection for maternal transmission enough to favour milk production by males in appropriate social settings. A caveat is that passive paracellular absorption of nutrients from the gut might also allow easier passage for toxins such as phyto-oestrogens into the host, in which case milk production would be pathological rather than adaptive[54,55].

Another open question is the effect of allonursing, where females other than the biological mother provide milk for the infant. This allows milk microbiomes to mix and should permit the spread of deleterious symbionts, just as it would if there was male lactation. Allonursing is rare in mammals, being documented in about 1–3% of extant mammalian species, with nursing of non-offspring being less than nursing of biological offspring in about 90% of these[56–58]. Transmission of harmful symbionts has yet to be considered among the costs of allonursing[59]. A low frequency of allonursing may retain a fitness threshold on symbionts near to that of strict maternal transmission, but a high frequency would not. So allonursing in mammals

**Table 1 | Matings with two host types, one type with symbiont community s and the other with community S augmented by a new symbiont, and the associated probabilities of the offspring's community, as explained in the text**

| mother | × | father | frequency | P(S) | P(s) |
|--------|---|--------|-----------|------|------|
| s | × | s | $q^2$ | 0 | 1 |
| S | × | s | $pq$ | $\alpha$ | $1-\alpha$ |
| s | × | S | $pq$ | $\beta$ | $1-\beta$ |
| S | × | S | $p^2$ | $\alpha+\beta-\alpha\beta$ | $(1-\alpha)(1-\beta)$ |

could be a natural laboratory for testing transmission of deleterious symbionts through milk.

Several further open issues about the theory should be kept in mind. Clearly the symbiont sieve operates for a limited period of time; after weaning it can only influence the dynamics of the gut microbiome through the imprint it leaves behind. In addition, the symbiont sieve is just one of several defences in mammary milk: bioactive molecules including oligosaccharides and immunoglobulins are also present, and help to protect the infant and influence the development of their gut microbiomes[60,61]. We have also not considered the detailed molecular pathways that control prolactin. These are complex, involving both epistatic interactions between genes and pleiotropic phenotypic effects that could constrain evolution[62]. While prolactin is often involved in regulation of lactation, high prolactin levels are recorded without lactation in male callitrichids (marmosets and tamarins), where paternal care is otherwise well developed[63].

Despite these open questions about lactation in mammals, it is notable how widespread maternal transmission of symbionts is in the natural world[35]. There is a danger of taking this for granted, rather than considering the drivers that lead to its evolution. It is, for instance, taken as given in the majority of host-microbiome models (see Methods: Relationship to other modelling frameworks). The symbiont sieve which comes with maternal transmission has not previously been considered, and offers a natural explanation. In the case of mammals, it gives insight into why male lactation is absent when other forms of paternal care are present, and into the role of vertically transmitted microbes during the early development of the infant gut microbiome.

## Methods
### Algebraic model
We suppose that the frequency of hosts carrying symbiont community $S$ (respectively $s$) in the parental generation is $p$ (respectively $q = 1 - p$). For generality, we write the probability that a female (respectively male) passes $S$ on to the next generation as $\alpha$ (respectively $\beta$). Thus, under biparental transmission $\alpha = 1, \beta = 1$, and under maternal transmission $\alpha = 1, \beta = 0$. $P(S)$ is the probability with which the offspring of the mating hosts carry the symbiont community $S$, and $P(s)$ is the probability with which the offspring carry the symbiont community $s$. The frequency of matings under random mating, and the resulting symbiont communities associated with these matings, are given in Table 1.

The frequency of hosts carrying $S$ in the next generation, $p'$, is given by the frequency of matings that pass $S$ on, multiplied by the fitness $w$ of hosts carrying $S$ relative to those those carrying $s$:

$$p' = \frac{w}{\bar{w}}\left(pq\alpha + pq\beta + p^2(\alpha+\beta-\alpha\beta)\right)$$
$$= \frac{pw}{\bar{w}}(\alpha+\beta-\alpha\beta p). \qquad (2)$$

Here the mean fitness of the population, $\bar{w}$, normalises the frequencies in the next generation so that they sum to 1. As this is for the next (rather than current) generation, $\bar{w}$ is the sum of the relative fitness of a community type multiplied by the frequency of new individuals with

that community type. Following Table 1, we have

$$\bar{w} = w\left(p^2(\alpha+\beta-\alpha\beta)+pq(\alpha+\beta)\right)$$
$$+ \left(p^2(1-\alpha)(1-\beta)+pq(2-\alpha-\beta)+q^2\right) \qquad (3)$$
$$= 1 - p(1-w)(\alpha+\beta-\alpha\beta p).$$

Taking $\phi = \alpha + \beta - \alpha\beta p$, the change in frequency of hosts carrying $S$ is then

$$\Delta p = p' - p$$
$$= \frac{pw}{\bar{w}}(\alpha+\beta-\alpha\beta p) - p \qquad (4)$$
$$= \frac{p}{\bar{w}}(\phi(p+wq) - 1).$$

The condition for the symbiont community $S$ to increase in the host population from one generation to the next is $\Delta p > 0$. Rearranging (4), the condition becomes $(p + wq) > 1/\phi$. When $S$ first appears in the host population, $p \approx 0$ and $q \approx 1$, so $S$ invades if $w > 1/\phi$. Fully maternal transmission of the symbionts ($\alpha = 1, \beta = 0$) implies $\phi = 1$, so $S$ can only invade if it gives a host fitness $w > 1$, i.e. a fitness greater than that of $s$. However, fully biparental transmission of symbionts ($\alpha = 1, \beta = 1$) implies $\phi = 2$, so $S$ invades if it gives a host fitness $w > 1/2$ relative to $s$, i.e. a fitness potentially lower than that of $s$. This demonstrates the reproductive boost that allows symbionts to invade under biparental transmission, even if they reduce host fitness. Finally, with $w < 1/2$, $S$ would not invade at all in this simple model, whether transmission is uniparental or biparental.

### Stochastic birth-death model
To investigate the dynamics of a host population with vertically transmitted symbionts, we constructed a continuous-time, stochastic, birth-death process with logistic density-dependence. This carries more information than the algebraic model above, including density dependence which allows a wider range of outcomes. The state of the host population at time $t$ consisted of the sex ♀, ♂ and the presence or absence $\{+, -\}$ of an added symbiont in each host individual $i = 1, ..., n(t)$, i.e.:

$$
\begin{array}{lll}
i_{\female}^+ : & \text{female} & \text{with symbiont set } S \\
i_{\male}^+ : & \text{male} & \text{with symbiont set } S \\
i_{\female}^- : & \text{female} & \text{with symbiont set } s \\
i_{\male}^- : & \text{male} & \text{with symbiont set } s.
\end{array}
$$

We write $n^+(t)$ as the number of hosts carrying the new symbiont (community $S$), $n^-(t)$ as the number without this symbiont (community $s$), and $n(t) = n^+(t) + n^-(t)$ as the number of hosts with or without the symbiont. Thus $n^+(t)$ gives a measure of the abundance of the additional symbiont in the host population at time $t$, and $n^+(t)/n(t)$ is its frequency in the host population.

The probability per unit time of death $d_i$ of host $i$ contained a logistic component common to all individuals, and a further dependence on symbiont status:

$$
\begin{array}{lll}
i_{\female}^+ : & d_i = \left(d_0 + d'n(t)\right)/w \\
i_{\male}^+ : & d_i = \left(d_0 + d'n(t)\right)/w \\
i_{\female}^- : & d_i = d_0 + d'n(t) \\
i_{\male}^- : & d_i = d_0 + d'n(t),
\end{array}
$$

where $d_0$ is an intrinsic death rate and $d'$ scales the density-dependent component of death. The death rate was modified by a factor $1/w$ in hosts carrying the additional symbiont (symbiont community $S$). This describes the effect of the new symbiont on its host: $w = 1$ is neutral, a beneficial symbiont ($w > 1$) lowers $d_i$ by a factor $1/w$, and a deleterious symbiont ($0 < w < 1$) raises $d_i$ by a factor $1/w$.

The per-female probability per unit time of giving birth $b_0$ was set to be independent of host population density. The sex of the newborn individual was assigned with an equal probability 0.5 to be female or male. Thus the probability per unit time with which a mother gave birth to a daughter, the key measure for host population growth, was $b_0/2$. Whether the symbiont was present (+) or absent (−) in a newborn host individual depended on the mode of vertical transmission:

> biparental : + if mother + or if father +
> − if mother − and if father−
> maternal : + if mother +
> − if mother − .

Mating was assumed to be at random, so a random father was chosen from the males present in the population for biparental transmission. This completes the specification of the stochastic, birth-death process.

We carried out realisations of the stochastic process using the Gillespie algorithm[64]. Simulated results were obtained with parameter values: $b_0 = 4$, $d_0 = 1$, $d' = 0.001$. The computations for Fig. 2 used a random value of $w$ drawn from a normal distribution with mean 1, and standard deviation 0.3 (truncated at 0 and 2.5). The initial number of host individuals was 1000 (the equilibrium point in the absence of the new symbiont), and the new symbiont was introduced to 10 individuals at the start, randomly distributed between females and males. Realisations were terminated when the symbionts were present in all hosts, or absent in all hosts, or the run-time had reached 200 time units. Five thousand independent realisations of each mode of vertical transmission were carried out. In almost all instances the outcome was presence in all hosts, or absence in all hosts. The four exceptions out of 10000 realisations were under maternal transmission, with $w$ close to neutrality.

We extended the stochastic process above to describe evolution of vertical transmission using two alternative genes at a locus on the Y chromosome. Male hosts were classified according to the gene they carried, $M^+$ allowing male transmission, and $M^-$ suppressing male transmission. Symbiont transmission from females was present throughout, so vertical transmission was biparental in crosses with $M^+$ males and maternal in crosses with $M^-$ males. There are four classes of males depending on transmission gene $\{M^+, M^-\}$ and presence or absence of the new symbiont $\{+, -\}$. To measure the association between transmission gene and symbiont status, we write the frequency of the classes in males as

> $p_1$ : frequency of $i_{M^-}^-$
> $p_2$ : frequency of $i_{M^-}^+$
> $p_3$ : frequency of $i_{M^+}^-$
> $p_4$ : frequency of $i_{M^+}^+$ ,

where $p_1 + p_2 + p_3 + p_4 = 1$. The coefficient of disequilibrium, $D = p_1 p_4 - p_2 p_3$, then measures the association between transmission gene and symbiont status. $D$ is positive if the new symbiont is under-represented in $M^-$ males and over-represented in $M^+$ males, and negative if vice versa. This completes the specification of the stochastic, birth-death process with evolution of vertical transmission. Computation of Fig. 4 was carried out with parameter values set to be the same as those in Fig. 2.

## Differential equation model

We note that a system of ordinary differential equations can be constructed for the mean behaviour of the stochastic process. We have used this to check the results of the stochastic realisations, to gain further understanding of the dynamics, and to check the robustness of the results to changes in parameter values.

In the absence of evolution (Figs. 2, 3, and 5a, b), the equations are:

$$\frac{dx^+}{dt} = \frac{b_0}{2} \frac{1}{x^+ + x^-} \left[ (x^+)^2 (\alpha + \beta - \alpha\beta) + (\alpha + \beta) x^- x^+ \right] - x^+ \left[ \frac{d_0 + 2d'\nu(x^+ + x^-)}{w} \right] + e_0 x^- , \quad (5)$$

$$\frac{dx^-}{dt} = \frac{b_0}{2} \frac{1}{x^+ + x^-} \left[ (x^-)^2 + (2 - \alpha - \beta) x^- x^+ + (1-\alpha)(1-\beta)(x^+)^2 \right] - x^- \left[ d_0 + 2d'\nu(x^+ + x^-) \right] - e_0 x^- , \quad (6)$$

assuming a 1:1 sex ratio to reduce the dimensionality of the system from four to two. The state variables are the density of female hosts $x^+$ with the symbiont community $S$, and the density $x^-$ with the community $s$. These state variables come from dividing the number of females with and without the additional symbiont ($n^+, n^-$) by a scaling parameter $\nu$ ($x^+ = n^+/\nu$, $x^- = n^-/\nu$). The mating classes are as defined in Table 1, the proportion of females (respectively males) passing community $S$ on to the next host generation being $\alpha$ (respectively $\beta$). We include here a term $e_0$, describing the per-capita rate at which hosts take up the new symbiont from the environment (i.e. horizontal transmission). Such transmission converts the host's symbiont community from $s$ to $S$. Under strict vertical transmission, $e_0 = 0$.

We used the differential equations to describe the dynamics under biparental transmission (setting $\alpha = 1$, $\beta = 1$), and under maternal transmission (setting $\alpha = 1$, $\beta = 0$). In the absence of horizontal transmission, the additional symbiont was added close to a boundary equilibrium point $I$ of the host population

$$\hat{x}_I^+ = 0$$
$$\hat{x}_I^- = \frac{1}{2d'\nu} \left( \frac{b_0}{2} - d_0 \right).$$

(Equilibrium points are calculated by setting the above system of differential equations equal to 0.) The initial per-capita rate of increase of hosts carrying the symbiont at this equilibrium point is

> biparental : $\left. \frac{1}{x^+} \frac{dx^+}{dt} \right|_I = b_0 \left( 1 - \frac{1}{2w} \right)$
> maternal : $\left. \frac{1}{x^+} \frac{dx^+}{dt} \right|_I = \frac{b_0}{2} \left( 1 - \frac{1}{w} \right)$,

giving a threshold fitness $w_0$ above which invasion of the symbiont happens with values $w_0 = 1/2$ for biparental transmission, and $w_0 = 1$ for maternal transmission. A second equilibrium point $II$ occurs at

$$\hat{x}_{II}^+ = \frac{1}{2d'\nu} \left( \frac{w b_0}{2} - d_0 \right)$$
$$\hat{x}_{II}^- = 0,$$

where every host carries the symbiont. This is unaffected by the mode of vertical transmission, but, unless the symbiont is neutral ($w = 1$), the symbiont changes the equilibrium host population size when it is present in all hosts. There is a threshold fitness $w_1$ at which the symbiont causes extinction of the host population ($\hat{x}_{II}^+ = 0$) at $w_1 = 2d_0/b_0$.

When horizontal transmission is absent, the thresholds for invasion by the symbiont $w_0$ and extinction of the host population $w_1$ are $w_0 = w_1 = 0.5$ under biparental transmission, and $w_0 = 1$, $w_1 = 0.5$ under maternal transmission, with the parameter values used here. Thus maternal transmission protects the host population from extinction, but biparental transmission could bring the host population size close to zero.

                                        

Numerical analysis was conducted using the above system of differential equations in Mathematica. We varied the parameters $w$ and $e_0$ between 0 and 1.5 in steps ranging from 0.001 to 0.1 and, at each combination of parameter values, the system was numerically solved and each physical ($x^+, x^- \geq 0$) solution's stability was calculated. This procedure was used to calculate the dividing lines between the regions in Fig. 5a and b, which were then verified using simulations. We then repeated this procedure for our robustness check for values of $b_0$ between 2 and 6 in steps of 0.25 and $d_0$ between 0.5 and 2 in steps of 0.125. This robustness check revealed that the majority of cases are as described in the main text with some differences as to the locations and curvature of the boundary lines. There is also the possibility of overlap between the regions of bimorphism and fixation, in which case the final state of the population depends on the initial conditions. As our focus in the main text is on invasions of either the new symbiont or maternal transmission, this would result in domination of the overlap by the bimorphic state (due to the small $x^+$ initial condition).

For Fig. 5c, we use our differential equation model with explicit types for each combination of sex, symbiont community and, for males, modifier gene (and thus transmission strategy). We represent the density of female hosts with the novel symbiont community as $x^+$ and the density of those without as $x^-$. Male host densities with the novel symbiont and with $M^-$ (uniparental, meaning $\alpha = 1, \beta = 0$) are $y^+$, without the novel symbiont and with $M^-$ are $y^-$, with the novel symbiont and with $M^+$ (biparental, $\alpha = 1, \beta = 1$) are $z^+$, and without the novel symbiont and with $M^+$ are $z^-$. We take $N = x^+ + y^+ + z^+ + x^- + y^- + z^-$. The resulting equations for the female hosts are then given by

$$\frac{dx^+}{dt} = \frac{b_0}{2}\left(\frac{x^+(y^+ + y^- + z^+ + z^-) + x^- z^+}{y^+ + y^- + z^+ + z^-}\right) - \frac{x^+(d_0 + d'\nu N)}{w} + e_0 x^- \quad (7)$$

$$\frac{dx^-}{dt} = \frac{b_0}{2}\left(\frac{x^- y^- + x^- y^+ + x^- z^-}{y^+ + y^- + z^+ + z^-}\right) - x^-(d_0 + d'\nu N) - e_0 x^-. \quad (8)$$

Meanwhile for male uniparental hosts we have

$$\frac{dy^+}{dt} = \frac{b_0}{2}\frac{x^+ y^+ + x^+ y^-}{x^+ + x^-} - \frac{y^+(d_0 + d'\nu N)}{w} + e_0 y^- \quad (9)$$

$$\frac{dy^-}{dt} = \frac{b_0}{2}\frac{x^- y^- + x^- y^+}{x^+ + x^-} - y^-(d_0 + d'\nu N) - e_0 y^-. \quad (10)$$

Finally, for male biparental hosts we have

$$\frac{dz^+}{dt} = \frac{b_0}{2}\frac{x^+ z^+ + x^+ z^- + x^- z^+}{x^+ + x^-} - \frac{z^+(d_0 + d'\nu N)}{w} + e_0 z^- \quad (11)$$

$$\frac{dz^-}{dt} = \frac{b_0}{2}\frac{x^- z^-}{x^+ + x^-} - z^-(d_0 + d'\nu N)) - e_0 z^-. \quad (12)$$

These equations are time integrated in order to give the smooth solutions present in Fig. 5c.

## Relation to other modelling frameworks

Developing a theory of vertical transmission requires recognition that both biparental and uniparental transmission could occur, and that they have strikingly different consequences for microbiota associated with the hosts. Future work will need to deal with the role the symbiont sieve plays in the makeup of the gut microbiome (not dealt with here). This calls for an understanding of the ecology of the microbial communities, moving on from neutral models[65] and those with frequency-independent selection[37,66], to take account of priority effects, succession and development of invasion resistance[67–69], which have measurable effects on host fitness[70]. Generalised Lotka-Volterra models[71,72] and consumer-resource models[73,74] are a step in this direction. In

addition, the feedback from the microbiome to host fitness can evidently drive host evolution, selecting host genes that control vertical transmission of symbionts. The key modelling challenge here is how to map properties of the microbiome (e.g. presence or absence of particular species and their abundances) to emergent fitness effects on hosts, which may depend on host genotype[38,75].

The evolutionary question we have tackled in this paper is quite different from a major current focus of host-microbiome theory: whether a symbiont evolves to increase host fitness at the expense of its own fitness within the microbiome[37]. Our question is how does the gut microbiome drive evolution of vertical transmission in mammalian hosts. Although this is just one of many points of contact between hosts and their microbiomes, it is especially important in setting the path along which the microbiome develops in the host offspring[76], with effects on lifelong health[39]. Our focus on evolution of vertical transmission also offers a contrasting perspective on horizontal transmission, over which hosts have relatively little direct control. Horizontal transmission is thought to decrease the potential for coevolution between hosts and their microbiomes[37]. However, our results show the presence of deleterious symbionts in the host population (sustained through horizontal transmission) actively holds in place the symbiont sieve through a continuing selection pressure against host modifier genes that would allow transmission from both parents.

### Reporting summary

Further information on research design is available in the Nature Portfolio Reporting Summary linked to this article.

## Data availability

Data generated from the archived code and the archived code are available at Github https://github.com/Brennen-Fagan/Maternal-Transmission, Figshare https://doi.org/10.6084/m9.figshare.22816529, and Zenodo https://doi.org/10.5281/zenodo.11208984. These source data, used to generate the Figs. 2– 5, are provided with this paper as well as Supplementary data 1[77]. The associated files can be found in the Supplementary data 1 folder within the corresponding figure's directory.

## Code availability

Mathematica and C code are also available at *Github* https://github.com/Brennen-Fagan/Maternal-Transmission[78], Figshare https://doi.org/10.6084/m9.figshare.22816529, *Zenodo* https://doi.org/10.5281/zenodo.11208984 and as Supplementary Code 1 with this paper. Supplementary Code 1 also mirrors the structure of the Supplementary data 1 in order to relate code with its outputs.

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

## Acknowledgements
B.T.F. is funded by a Leverhulme Trust Research Centre—the Leverhulme Centre for Anthropocene Biodiversity, grant number RC-2018-021. We thank the University of York Reading Group on Stability and Complexity in which discussions on the dynamics of gut microbiomes led to this work, J. A. Fagan for useful discussions, B. König for comments on allonursing, and V. Hutson, C. M. Law and J. W. Pitchford for comments on the paper.

## Author contributions
All authors, B.T.F., G.W.A.C. and R.L., contributed equally.

## Competing interests
The authors declare no competing interests.
