## [Peer Review File · Nature Communications]

Maternal transmission as a microbial symbiont sieve, and the absence of lactation in male mammalsReviewers' Comments:

Reviewer #1:

Remarks to the Author:

This paper puts forward the hypothesis that vertical transmission of microbiota via mammary milk is another possible reason for the absence of male lactation. This hypothesis is underpinned by a simple mathematical model. The question the study aims to tackle is very interesting and the writing of the paper is very well and clear. As far as I can tell, the results and conclusions are sound within the scope of the assumptions of the model.

I feel, however, ambivalent about the model (or models) and the hypothesis it intends to support. On the one hand simple models are obviously great, and under its specific set of assumptions this one makes a good point, which boils down to: "if the probability of vertically transmitting a deleterious symbiont is reduced by removing one of the vertical transmission pathways, the symbiont will not spread as easily, making the restriction of pathways adaptive". But due to its deliberate simplicity, the model can also not say much more than this and as such I am not sure how much insight it really provides.

A case in point is the fact that, as the authors themselves point out, there are potentially some rare exceptions to maternal-only lactation. These rare exceptions to the general rule seem to me to be very interesting, but as far as I can see the model can not really say anything about these exceptions and under which circumstances we should expect to see them, if at all. There is a speculative paragraph about this in the discussion (Dayak fruit bats), but this is where a model could actually provide some more concrete insights. Especially since, as far as I am aware, it is still unclear whether the observed male lactation in fruit bats is actually adaptive or, rather, pathological.

Specifically, high microbiome turnover is mentioned as one of the possible reasons contributing to the fruit bat exception, but the model makes the very restrictive assumption that the microbial community (with or without deleterious symbiont) is fixed for the entire life span (or at least the reproductive span) of the host. But often microbiomes are found to be very transient and variable, especially during the first stages of host development, so that deleterious symbionts may well be lost by the time the host reproduces, in particular once the immune system kicks in. The potential for (rapid) microbiome turnover and the loss of deleterious symbionts after initial vertical transmission would seem to severely restrict the conditions under which the "symbiont sieve" would be relevant. I would like to see the implications of the "fixed community" assumption at least discussed somewhere in the paper.

Another strong assumption is that there is no fitness effect at all of varying milk supply for the offspring, especially no positive fitness effect of increased supply by biparental lactation, which may then trade-off with the risk of vertically transmitting deleterious symbionts. And vice versa, since there is no negative effect of decreased milk supply, wouldn't the current model lead to the non-sensical conclusion that no lactation at all is the best course of action for parents (which the model doesn't allow for of course)? This should at least also be discussed.

Overall I am left with the feeling that this paper puts forward an overly simplistic model for a very complex problem, which makes it hard to judge its real-world relevance and significance. Nevertheless, if at least the implicit assumptions mentioned above (fixed microbial communities and no fitness effect of varying milk supply) are stated and discussed more clearly, I don't see a reason for not putting the hypothesis out there via *Nat. Commun.*

Other comments:

If I understand correctly, in the discussion the authors seem to imply that the absence of "widespread deleterious symbionts" in mammary milk would somehow constitute a test of the theory. But of course such "symbionts" could be absent from mammary milk for any number of reasons, not least the

specific composition of mammary milk. As such this is no evidence for or direct test of the theory, it merely shows that the model is not inconsistent with the known facts about mammary milk, i.e. that disease transmission through breast feeding is in fact rare. This is a much lower bar and the corresponding paragraph about testability should be rephrased accordingly.

The basic premise of the model, i.e. reduced probability of vertical transmission of deleterious symbionts by restricting vertical transmission pathways, probably also applies to other forms of vertical transmission? Shouldn't this then also select against other forms of biparental care with possibility of transmission, several of which nevertheless exist? Can the authors comment on this and potentially discuss it in the paper?

The mentioned conditions suppressing selection against biparental transmission (i.e. high horizontal transmission and rapid microbiome turnover) probably apply to many more species other than fruit bats, but only(?) the fruit bats are known (potential) exceptions to the general rule of uni-parental lactation. Wouldn't this suggest that there is a stronger, more general, selection pressure against male lactation other than vertical transmission of deleterious symbionts? Or even a basic physiological reason making male lactation rare/unlikely?

Fig. 5

Provide meaning of the different regions (e.g. "host bimorphic", "host extinction", "symbiont fixation") directly in the figure, rather than labeling them "I, II, III" and referring to the labels in the captions.

Reviewer #2:

Remarks to the Author:

It is a very theoretical and speculative study that is not supported by any experimental validation. In fact, despite the large use of mathematical models aimed to interpret biological phenomena this study is totally lacking biological experiments corroborating the results predicted by the mathematical algorithms here described. This represents a very serious limit of this study and make the overall manuscript very notional. Somehow this manuscript resembles more a review more than an original paper. The introduction section is very verbose and does not provide a clear state of the art about the topic of this study. In contrast, it is very rich in author interpretation/speculation about the absence of male lactation. The results section is difficult to follow to a non-expert in mathematical model and again is very rich in speculative sentences that are not supported by any experimental data (e.g., metagenomic data about human gut microbiota).

Reviewer #3:

Remarks to the Author:

I liked this paper a lot, and I enjoyed reading your new and innovative ideas. One of the major strengths of the paper is also something that may pose some problems. The article essentially provides a new framework by which to better understand parental investment differences in infants, without having to rely on existing sexual selection hypothesis to justify why females invest more and males generally invest less. I like this, and I think we are overdue for some fresh perspectives on the unequal investment in offspring by different sexes. The authors do briefly mention Trivers parental investment theory, and mention that their work adds to this paradigm, so it is not in opposition to it. That's fine but give us the details on this and expand what you mean. I think the authors need to better situate their theory within the existing paradigm of parental investment and sexual selection theories, in both the introduction and in the discussion. This will make the main ideas put forward even more relevant and more clearly influential for the evolutionary biology community. Not making this more explicit may make it difficult for your theory to be immediately accepted alongside the existing frameworks. People can speculate about how your ideas fit in with major paradigms in

evolutionary biology but might as well just do the work for them, and tell us exactly how framework fits within the existing paradigm of sexual selection driving differences between males and females in parental investment. I think doing this will help your paper have an even bigger impact on the biological community!

Another issue in the paper is the consideration of prolactin, as it often can be very high in males of cooperative breeding species, or where paternal care is high. And yet males still do not lactate. So the consideration of prolactin levels to explain the lack of lactation needs to be fleshed out a bit better, given what we currently know. I provided several comments on this throughout the paper, so hopefully that will give you some ideas on how to better explain the mechanism behind the lack of lactation in males... I mentioned the possibility of the interaction between physiology and hormones, although I know this idea might be a anthropomorphism since we expect many females to grow mammary gland tissue but not males. Not sure about differences in breast/chest physiology between males and females in other mammals.

Lastly, some of the language can be adjusted to be more inclusive. Natural birth could be changed to vaginal birth. Breast feeding could be changed to breast and chest feeding.

See my other detailed comments below:

INTRODUCTION:

2nd sentence in intro: change "which" to "that"

Genetically male mammals... Could it be genetically and phenotypically male mammals? I am trying to think of situations where you can have genetic males that are phenotypically female (i.e. intersex).

Another thing that I think needs to be mentioned in the intro around the discussion of monogamy and paternity certainty, is that even in monogamous or polyandrous species, like in callitrichids, where males provide a lot of care, prolactin levels do increase a lot in these males, right when the female become pregnant, and they remain elevated for these males throughout the pregnancy and while the infants are small. The prolactin inhibits testosterone and helps induce paternal/caregiving behavior to the female and infant by the male. But if these males/fathers have higher prolactin, how come they are not lactating? Their prolactin levels are not high enough to induce lactation? So, the mechanism of higher prolactin is there too for the males in the monogamous species but it does not lead to lactation and nursing of young. Could be useful to add and consider because I think it would further support your point, that there seems to be a selection pressure against lactation by males, even in species that have all of the behavioral and hormonal factors set in place that would make male lactation possible. They still don't do it!

I just checked my PowerPoints because I teach this concept in one of my classes. The fathers in callitrichids actually have prolactin levels that are equally as high as the mothers. Also, more experienced fathers have higher prolactin levels than less experienced fathers. Sorry I can't provide you with the actual references because I can't remember them...

"restoration of normal prolactin levels". What is normal? Implying lactation is not normal... Change this sentence to be more neutral. Maybe "reversible once prolactin levels decrease again after lactation for the current offspring ends".

such elements is confusing. What elements. Be specific again: gut microbiome elements?

When you talk about vertical transmission of gut microbiome elements, I have some questions. What is vertical transmission? Define it. Why not just transmission? Is it because it is not horizontal transmission? So I guess explain vertical transmission enough to understand why you use that term. Also, vertical transmission of the gut microbiome... who's gut microbiome? From the parent/adult to

the infant? Make clear.

Reading the rest of the paper, I now think you should certainly define in the intro your terms of vertical versus horizontal transmission.

Same idea for the next several sentences where you talk about the host? You mean the host population of the infant/offspring, so the microbes that the infant is already developed in-utero? Also, what are symbionts? Maybe this term is regular for most people but some may read this and not quite know... is it different just from the term symbiote?

"switched off", "has remained switched off"- is a bit odd... Is there an alternative that doesn't sound as if we're talking about a light switch or turning a machine on and off? Maybe the word "suppressed" or "keep male lactation from happening", or "from being expressed", etc.

"do impair male activity"- could you be more specific as to what male activity you are referring to? Be specific here as to what male reproductive activity you are talking about? Also maybe say why? Prolactin inhibits testosterone which blah blah? Just an example, I do not know the mechanism. infants born naturally, or natural birth needs to be changed. You mean infants born vaginally, or infants that exit through the birth canal.

RESULTS AND METHODS:

I cannot comment on any of the algebraic models because I am far from a math expert (I am a biological anthropologist). The explanations of the models seem good/make sense from the results section explanations, although I admit I sometimes lost track of the reasoning, which is mostly because this level of math is difficult to assimilate for someone who doesn't use it often.

I hope another reviewer will be able to comment in detail on the logic of the algebraic models. Sorry, it's beyond my strengths

DISCUSSION:

At times, it is unclear if you are talking about one of the parents or the infant/offspring. Can you try and make clearer throughout. For instance, in this sentence:

"We cannot prove that the danger of invasion by deleterious elements in the gut microbiome is the reason why there is no male lactation in mammals, so we put forward this argument as a hypothesis.

Can you add either "infant" or "offspring" before "gut microbiome".

Low prolactin level in males may be the case for most species, but again, doesn't seem to be the case for some cooperative breeding species like the callitrichids, or even humans, where fathers show high prolactin. Thus, in the discussion, I am not sure if genes for low prolactin levels in males is the best way to explain the absence of lactation in males.

Careful that sometimes you use word "which" when it should be the word "that". Example, for the phrase: "operating through host modifier genes which control these levels" should be "that", not "which".

Unruly mob of microbes is delightful. I like it but is it too casual/ jargon-y? If nobody else has a problem with it, I am fine with it too.

If you wanted to be more inclusive, you could say breast and chest feeding when referring to breastfeeding in humans... Although some might argue that this is not necessary. I like to say it that

way when possible, but up to you authors.

The fruit bat example is very interesting. I was also thinking that in humans, it is possible to make male lactate, isn't it, with domperidone to increase prolactin levels a lot? I would also say transgender females, but I guess they would have been exposed most likely artificially to progesterone and estrogens hormone replacement so that would affect chest/breast physiology ahead of lactation.... Hmm. But with Cis males, it is just that with a cis male physiology, one would need prolactin levels that are so much higher than for lactating females, just to induce milk synthesis in male glands? So maybe the idea of prolactin is more about the interaction between physiology and hormones? I don't know. Maybe that's too much to consider in this paper... Certainly your ideas about prolactin genes and co-evolution with the symbionts are interesting but details need to be ironed out a bit.

Reviewer: Iulia Badescu

Reviewer #4:

Remarks to the Author:

We enjoyed reviewing this thought-provoking manuscript that brings together evolutionary theory, the microbiome, lactation, and mathematical modeling. We believe that the authors have put forth an intriguing hypothesis, but we have several concerns about the approaches and frameworks used in the paper. As detailed below, we are concerned that some of the models' assumptions are not well supported, and we believe that the manuscript lacks definitions of multiple key terms. We provide more detailed points below.

1. We are unsure whether the goal is to explain the absence of male lactation in humans specifically, or in mammals more generally. The human microbiome is different from other mammals and different still from the proto/early-mammals in which lactation first evolved, making this distinction important. Relatedly, the manuscript is also lacking in certain components of the broader evolutionary context.

2. The manuscript would benefit from clear definitions in the Introduction, as this would help the reader better understand the framework and approach. Definitions are needed for:

- a. Symbiont – is any microbe in milk automatically considered a symbiont? How do the authors delineate between related ecological relationships like symbiosis, commensalism, and mutualism?
- b. Vertical and horizontal transmission – it seems that the authors intend to use 'vertical transmission' to mean any microbial transmission from parent to offspring. We question this usage, since microbes that are transmitted from parents to offspring are not passed down through the same pathways as inherited genes are. Instead, one could argue that all maternal-offspring microbial transmission is horizontal and/or environmental— since transmission is thought to start at birth (and not in utero, as the authors note), then one could perceive microbial transmission to occur within the 'birth environment,' just as milk microbes could be shared through the 'feeding environment.' Moreover, if the authors intend to use vertical transmission to mean parent-offspring transmission, then they should also be concerned with microbial transmission that occurs via skin-to-skin contact or the exchange of oral microbes. In other words, there are many routes of microbial transmission to offspring beyond milk, including behavioral routes that, like breastfeeding, have likely been shaped by natural selection (e.g. alloparental/cooperative breeding; sociality and group living). If the authors want to focus on vertical transmission, it needs to be clearly defined and supported.
- c. Deleterious microbe – does this term apply to all types of microorganisms? The authors briefly mention viruses, but studies of the milk microbiome are often focused on bacteria. It is not clear which types of microbes, deleterious or otherwise, are most likely to be transmitted under the pathways and models presented here. We are concerned about the binary delineation between "good" and "bad" microbes, which often does not apply to complex microbial communities.

3. Relatedly, we also question why the authors chose to exclude host-to-host horizontal transmission.

The social environment is a known pathway of microbial sharing across myriad host species (see recent studies on lemurs, howler monkeys, baboons, chimpanzees, and humans, as well as birds and bees), and in the context of this paper, we consider the social environment as a critical source of potential pathogen exposure for offspring. Further, it is not clear to us why the authors assume that milk is such a critical source of potential 'deleterious' microbial transmission, when it is well known that other routes of pathogen sharing, including those from conspecifics, pose considerable health risks to hosts. Given the evolutionary framing of this manuscript, the omission of well-known selection pressures that stem from socially transmitted pathogens is concerning.

4. This paper makes a fundamental adaptationist assumption: that the absence of a feature is evidence of selection against the development of that feature. It must be considered that the presence or absence of a feature is not necessarily adaptive, but may simply be unobtrusive enough to not be shaped by natural selection. Lactation and gestation are calorically expensive processes and must be worth the price to develop. Lactation first evolved in species where males had no role in the care or protection of infants, and it continued to evolve absent paternal care. The proposed selection mechanisms took place when the microbic composition of mammalian guts was very different than in humans today. To this day, paternal care—and particularly calorically expensive paternal care—is the exception, not the rule. Intensive parental care as seen in humans and in some primates is a recent evolutionary development, and reconfiguring a 250-300 million year-old system may not be feasible for a species like humans that has employed the cooperative care model for far less time.

a. Additionally, in its own way, cooperative care supplements maternal caloric investment in the form of infant carrying and cosleeping. (See the work of Lee Gettler on male carrying and cosleeping)

5. If the focus is to be on humans, this paper may be strengthened by framing it within the wider question of why humans (and some primates) do not seem to share the resource burden of lactation by employing allonursing as part of the cooperative care model.

a. In primates, evidence on allomaternal nursing and energetics is mixed. One study found that it is more likely to occur among cooperative breeders such as tufted capuchin monkeys (Hewlett and Winn 2014 - "Allomaternal Nursing in Humans" <https://doi.org/10.1086/675657>), but another found that it was more likely to occur when costs were lower—suggesting physiological constraints (MacLeod and Lukas 2014). Limited evidence suggests that although allomaternal nursing is widespread in humans, it is also rarely done—typically only in emergency situations, and typically only between kin. This begs the question: if it is rare, why?

b. One observation that was made was that with the (limited) cross-cultural data was that allomaternal nursing was more common in cultures that lived in tropical climates, where infectious disease burden is higher, particularly parasitic infection (See Hewlett and Winn, 2014).

c. Another observation (again, from limited data) was that the allonursers were slightly more likely to be paternal kin than maternal. This is interesting considering the increasingly demonstrated epigenetic and microchimeric effects of lactation.

6. In the discussion of male lactation in humans, this paper would benefit from a more in-depth discussion about alternative explanations for the absence of widespread lactation in males, and life history theory and cooperative care/biocultural care models. In humans, and to a degree in higher apes, lactation is intensely social and behavioral, and so biocultural contexts must be considered in explaining the presence or absence of a behavior, or of a developed mammary gland. Gestation and lactation are energetically expensive for the maternal body (particularly in humans), which would perhaps make male lactation beneficial regardless of the presence of deleterious microbic species, but at a cost to the male. In humans, that cost includes sharing the expensive caloric burden of an infant with a rapidly growing brain, and the homeostatic regulation provided by the skin-to-skin contact associated with breastfeeding. (See the work of Bogin, Bragg, and Kuzawa 2014 doi: 10.3109/03014460.2014.923938 and other recent work on cooperative care in the human lineage and non-human primates)

a. Additionally, lactational amenorrhea induced by breastfeeding (affecting only the female) leads to the wider interbirth intervals (compared to other primates) necessary to support the extended and

calorically expensive early growth period characteristic of higher apes.

7. While we understand that all models are built on a set of assumptions, we are concerned about the assumptions used as the basis of this analysis. In particular, the first assumption (the fate of the new symbiont is independent of the resident microbiome) violates fundamental principles of microbial community ecology (including the ecology of milk components and bacteria interacting within the infant gut). As supported both theoretically and empirically, microbial communities are inherently compositional, dependent networks. In fact, new bioinformatics and statistical approaches have been recently developed to address these core properties of microbial communities. As such, we question the validity and applicability of a model that is built on a violation of these well-known properties of microbial communities.

8. The paper is lacking in a discussion of the immune system, which has an intimate relationship to the microbiome. In fact, it has been posited that the genetic component of immunity may be a mechanism for the 'inheritance' of a microbiome. Lactation also confers immune benefits more generally. This is why, early in the COVID-19 pandemic, breastfeeding was recommended by the WHO even as very little was known about the virus and vertical transmission from mother to infant. A fundamental assumption for the authors' hypothesis is that the selective pressure exerted by the milk microbiome would have to surpass the other adaptive components of lactation, such as the profound immune benefits and the aforementioned defraying of caloric costs. We are not convinced that it does.

9. A deeper discussion of lactation physiology would also be of benefit. The hormonal changes associated with pregnancy are essential to inducing the proliferation of lobules in the mammary gland that make milk synthesis possible. Although lactation without pregnancy is possible in humans, it is difficult to generate a full supply, and success is positively associated with gravidity. (See also: grandmother hypothesis and induced lactation/adoptive lactation) This suggests there have been evolutionary constraints operating on male mammary tissue development. It is also worth noting briefly in your paper, as not all readers may be aware, that males do in fact have some mammary tissue, and can develop breast cancer; their mammary tissue just doesn't develop in adolescence the way female mammary tissue does.

10. "Socially monogamous mammals" (Page 1, paragraph 2) – this is only about 3% of mammals. Lifelong pair bonding is relatively rare in mammals, and primates (including humans) are not more monogamous than other mammals. In highly social species, including humans, up to 20% of offspring are conceived as a result of copulations outside of the primary pair bond (See Reichard, U. H., and C. Boesch 2003. *Monogamy: Mating strategies And partnerships In birds, Humans and Other Mammals*. Cambridge University Press.)

11. "Gut microbiomes of infants born naturally" (Page 2, paragraph 3) – we strongly suggest using "vaginally" instead of "naturally." "Natural" is unspecific and qualifying. It is mostly the vaginal microbes that are of interest here, but it is notable that mode of birth is not the only aspect of birth which may affect microbial transmission; others include the absence or presence of skin-to-skin practices, the surface and air microbial environment in hospitals, and the use of antibiotics.

12. Phyllosymbiosis (page 6): there is conflicting evidence for the strength of this phenomenon across clades. We encourage the authors to review the literature and note any limitations.

13. The first sentence of the Discussion uses very strong, dramatic language. We suggest toning this down this sentence.

Responses to Reviewer Comments

We are very grateful to the reviewers for taking the time to provide such detailed and insightful comments; these have helped us improve the paper on many counts.

Below we have responded to each reviewer comment in detail, as well as providing a version of the manuscript with all changes highlighted for clarity.

Reviewer #1 (Remarks to the Author):

This paper puts forward the hypothesis that vertical transmission of microbiota via mammary milk is another possible reason for the absence of male lactation. This hypothesis is underpinned by a simple mathematical model. The question the study aims to tackle is very interesting and the writing of the paper is very well and clear. As far as I can tell, the results and conclusions are sound within the scope of the assumptions of the model.

I feel, however, ambivalent about the model (or models) and the hypothesis it intends to support. On the one hand simple models are obviously great, and under its specific set of assumptions this one makes a good point, which boils down to: "if the probability of vertically transmitting a deleterious symbiont is reduced by removing one of the vertical transmission pathways, the symbiont will not spread as easily, making the restriction of pathways adaptive". But due to its deliberate simplicity, the model can also not say much more than this and as such I am not sure how much insight it really provides.

Authors' Reply: We understand Reviewer #1's concern about the simplicity of the basic argument, and are ourselves surprised that the symbiont sieve is not a standard part of thinking about vertical transmission of symbionts. Nonetheless, even this simple point sheds new light on why lactation in male mammals could be problematic. We think a good starting point is to make the basic argument as simple and clear as possible, but appreciate that the work does need to take account of the real complexity of gut microbiomes in due course.

Responding to this and also to points 4.2c and 4.7 of Reviewers #4, we have taken a step towards real-world complexities in revising the paper. We now show the symbiont sieve operating on a microbiome with 20 symbiont taxa (as opposed to a single taxon). We have taken a mean value of the 'w's of the symbiont taxa as the effect of the microbiome on host fitness. This is sufficient to show how important the symbiont sieve is for transmission of the microbiome as a whole. To make room for this we have replaced the original Figure 3 with some results on the multispecies microbiome, and replaced the original text on Figure 3 with new text. We have tested the symbiont sieve with other numbers of symbiont taxa (not shown), and our results suggest the sieve works with an arbitrary number of symbionts. We think the new results strengthen the argument for the importance of maternal transmission of microbiomes.

(Comment 1.1) A case in point is the fact that, as the authors themselves point out, there are potentially some rare exceptions to maternal-only lactation. These rare exceptions to the

general rule seem to me to be very interesting, but as far as I can see the model can not really say anything about these exceptions and under which circumstances we should expect to see them, if at all. There is a speculative paragraph about this in the discussion (Dayak fruit bats), but this is where a model could actually provide some more concrete insights. Especially since, as far as I am aware, it is still unclear whether the observed male lactation in fruit bats is actually adaptive or, rather, pathological.

Authors' Reply: Reviewer #1 is correct here and we could have communicated this better. Our reference to the Dayak fruit bat in the Introduction was simply to provide evidence that it is possible for male mammals to produce milk. We have changed 'lactation' to 'milk production' in the context of the Dayak fruit bat in the first paragraph of the paper, and in paragraph 7 of the Discussion, to allow for the possibility that this a pathology from phyto-oestrogens in their diet (Racey et al. 2009, TREE 24:354) as the reviewer points out.

To improve the discussion of the Dayak fruit bat, we have changed paragraph 7 of the Discussion. We now note that there is strong evidence that mammals and birds with the capacity for flight have gut microbiome structures different from those that do not fly; this comes from the meta-analysis of 315 mammal and 491 bird species (Song et al. 2020). Song et al. (2020) make what is to us a convincing case for a suite of gut traits expected to emerge under the constraints of flight in mammals and birds, including low body mass of which a short gut is part, more paracellular absorption of nutrients compensating for a small gut, less anaerobic conditions in the gut allowing more uptake of microbes from the environment, and less overall dependence on a gut microbiome. Given social conditions under which there is a benefit from male lactation, coupled to a low cost from biparental transmission of the gut microbiome, the balance of selection pressures might allow milk production by males. This would be consistent with our model, and we think it is worth suggesting, since Dayak fruit bats are a strikingly unexplained counter-example among mammals. However, we also appreciate that paracellular absorption of nutrients might allow phyto-oestrogens to be taken up, giving rise to male milk production, so we have now added a warning about this.

(Comment 1.2) Specifically, high microbiome turnover is mentioned as one of the possible reasons contributing to the fruit bat exception, but the model makes the very restrictive assumption that the microbial community (with or without deleterious symbiont) is fixed for the entire life span (or at least the reproductive span) of the host. But often microbiomes are found to be very transient and variable, especially during the first stages of host development, so that deleterious symbionts may well be lost by the time the host reproduces, in particular once the immune system kicks in. The potential for (rapid) microbiome turnover and the loss of deleterious symbionts after initial vertical transmission would seem to severely restrict the conditions under which the "symbiont sieve" would be relevant. I would like to see the implications of the "fixed community" assumption at least discussed somewhere in the paper.

Authors' Reply: It was not our intention to suggest that the gut microbiome is fixed for the entire life span. Naturally, we agree that it is open to microbes throughout the life of the host, and vertical transmission deals just with one (important) path of entry, acting near the start of life. Our section on horizontal transmission recognises this. However, there is evidence that certain elements of the gut microbiome are relatively fixed. For example, Moeller et al. (2018) found individual-specific gut microbiota compositions of wild mice were maintained in descendants over multiple generations.

Mammary milk is an important source of microorganisms in the early stages of assembly of the gut microbiome: current evidence suggests approximately 10^3 to 10^4 colony-forming units per ml of milk in healthy human females (Jeurink et al. 2013). It is also important to keep in mind that the gut is anaerobic and unsuitable to many of the aerobic microorganisms living in the external environment.

To improve our presentation of this topic, we have rewritten paragraph 4 of our Introduction. In doing so, we have stressed that mammary milk is only one of a number of sources from which the microbiome of the infant gut is assembled. We have referenced Jeurink et al. (2013), and commented on the paper by Moeller et al. (2018). We have also included a caveat in paragraph 9 of the Discussion that, after weaning, the milk microbiome can only affect the subsequent dynamics of the gut microbiome through the imprint it has left on the assembly process. We have also pointed out near the start of the section on horizontal transmission that this is, in effect, a first step beyond the assumption of a symbiont community affixed near the time of birth.

(Comment 1.3) Another strong assumption is that there is no fitness effect at all of varying milk supply for the offspring, especially no positive fitness effect of increased supply by biparental lactation, which may then trade-off with the risk of vertically transmitting deleterious symbionts. And vice versa, since there is no negative effect of decreased milk supply, wouldn't the current model lead to the non-sensical conclusion that no lactation at all is the best course of action for parents (which the model doesn't allow for of course)? This should at least also be discussed.

Authors' Reply: The reviewer is right to note that our focus is on the costs of biparental lactation, rather than the benefits. In particular we were interested in cases where the benefits of biparental care appear sufficiently high that such care is selected for (as with Azara's owl monkeys), and yet lactation is not. This led to us keeping a keen eye on costs to male lactation that are not accounted for in the literature.

That said, we do agree that accounting for benefits of male lactation for progeny is an important consideration. We note that such benefits are relatively straightforward to account for within our modelling framework. Drawing on Kokko and Jennions (2008), we can introduce an additional survival probability on infants that is a function of investment in maternal lactation (α) and paternal lactation (β). The survival probability is given by $\exp[-\gamma/(\alpha+\beta)]$, where decreasing γ decreases the relative benefits of male lactation (see below, left panel). Our preliminary results show a sharp transition at which biparental lactation can invade (see below, right panel).

We believe this represents a very interesting research avenue. However as even the simplest model evolutionary model of this system (with just two symbiont communities) consists of six ordinary differential equations (for two females classes for each symbiont community, and similarly two biparental males and two uniparental males), we do not have the space to investigate this within the scope of the current paper.

(Comment 1.4) Overall I am left with the feeling that this paper puts forward an overly simplistic model for a very complex problem, which makes it hard to judge its real-world relevance and significance. Nevertheless, if at least the implicit assumptions mentioned above (fixed microbial communities and no fitness effect of varying milk supply) are stated and discussed more clearly, I don't see a reason for not putting the hypothesis out there via Nat. Commun.

Authors' Reply: We are glad that this reviewer sees the hypothesis as being suitable, in principle, for publication in Nature Communications. We have now taken two steps towards the complexities raised in the review. First, we have further generated results which suggest the symbiont sieve operates on a microbiome with an arbitrary number of interacting symbionts. Secondly, we have shown that there is a region of parameter space in which the symbiont sieve operates when benefits of male lactation are taken into account. The model at this stage remains simple, but we believe there is great potential for developing an active research programme from these results.

Other comments:

(Comment 1.5) If I understand correctly, in the discussion the authors seem to imply that the absence of "widespread deleterious symbionts" in mammary milk would somehow constitute a test of the theory. But of course such "symbionts" could be absent from mammary milk for any number of reasons, not least the specific composition of mammary milk. As such this is no evidence for or direct test of the theory, it merely shows that the model is not inconsistent with the known facts about mammary milk, i.e. that disease transmission through breast feeding is in fact rare. This is a much lower bar and the corresponding paragraph about testability should be rephrased accordingly.

Authors' reply: We appreciate Reviewer #1's concerns here. We meant testable in the (Popperian) sense of falsifiable and have now made this clear in the second paragraph of the Discussion.

(Comment 1.6) The basic premise of the model, i.e. reduced probability of vertical transmission of deleterious symbionts by restricting vertical transmission pathways, probably also applies to other forms of vertical transmission? Shouldn't this then also select against other forms of biparental care with possibility of transmission, several of which nevertheless exist? Can the authors comment on this and potentially discuss it in the paper?

Authors' reply: In principle, we agree with Reviewer #1.

If Reviewer #1 means vertical transmission in other taxa, it is notable that maternal transmission of symbionts appears to be a widespread organising principle among organisms with sexual reproduction, referred to as 'Mom knows best' by Funkhouser and Bordenstein (2013). The arguments developed here have enough generality to

give some insight into why this organising principle exists, and we have added a sentence at the end of the Introduction making this point. Within the context of the gut microbiome, we have not seen evidence of other types of biparental care making a contribution comparable to the milk microbiome.

If Reviewer #1 means vertical transmission of other microbiomes in a single taxon, e.g. the skin microbiome, such a microbiome has direct external exposure to the environment and would seem to be in the very high horizontal transmission region of Figure 5 ($e_0 > 1$). In this region, the amount of horizontal transmission would render the mode of vertical transmission neutral with respect to the microbiome (although not to other pathways to increased fitness). This is an interesting point, and we have alluded to it in the paper (second to last paragraph before the discussion).

(Comment 1.7) The mentioned conditions suppressing selection against biparental transmission (i.e. high horizontal transmission and rapid microbiome turnover) probably apply to many more species other than fruit bats, but only(?) the fruit bats are known (potential) exceptions to the general rule of uni-parental lactation. Wouldn't this suggest that there is a stronger, more general, selection pressure against male lactation other than vertical transmission of deleterious symbionts? Or even a basic physiological reason making male lactation rare/unlikely?

Authors' reply: The conditions that might relax the full force of selection against biparental transmission appear to be associated with flight. The evidence from phyllosymbiosis is that gut microbiomes of bats have a lot more in common with those of birds which fly, than with other mammals (Song et al. 2020, see also response to Comment 1.1). We speculate that the role of the gut microbiome might be reduced enough to create conditions in which biparental feeding of offspring, which is widespread in birds, becomes possible. This is just a speculation, and the reviewer rightly asks, if it happens in fruit bats, why not in other taxa? Interpreting this as other 'bat' taxa, a possible reason is that, although the costs from biparental transmission are reduced, this reduction is still barely enough for biparental feeding to be advantageous for Dayak fruit bats. We think the speculation is worth making because male milk production in Dayak fruit bats is an exception in mammals, and there is currently no explanation other than that it is a pathology.

Fig. 5

(Comment 1.8) Provide meaning of the different regions (e.g. "host bimorphic", "host extinction", "symbiont fixation") directly in the figure, rather than labeling them "I, II, III" and referring to the labels in the captions.

Authors' reply: We have done this, and updated the legend to Fig 5 and text accordingly.

Reviewer #2 (Remarks to the Author):

(Comment 2.1) It is a very theoretical and speculative study that is not supported by any experimental validation. In fact, despite the large use of mathematical models aimed to interpret biological phenomena this study is totally lacking biological experiments corroborating the results predicted by the mathematical algorithms here described. This represents a very serious limit of this study and make the overall manuscript very notional...

Authors' reply: We thank you for your remarks and candour. We naturally agree that controlled biological experiments would be ideal, and hope our theoretical investigation will encourage studies in this area. However even in the absence of experiments, much can be learned from existing empirical observations, which we have been careful to draw upon. More broadly we would expect further theoretical developments would be helpful in designing appropriate biological experiments that can interrogate our hypothesis. We note that the role of theory in evolutionary biology is extremely important and helps in this regard (see, for instance, National Research Council (2008), Servedio et al. (2014), and Longo and Soto (2016)).

(Comment 2.2) ... Somehow this manuscript resembles more a review more than an original paper. The introduction section is very verbose and does not provide a clear state of the art about the topic of this study. In contrast, it is very rich in author interpretation/speculation about the absence of male lactation...

Authors' reply: As we are attempting to introduce a new hypothesis into the literature, we felt it necessary to simultaneously discuss the nature of the field on that front (and thus "resemble a review") while also illustrating where our hypothesis fits in ("rich in author interpretation/speculation"). While we have tried to make our introduction and discussion more precise on these points, we also note that we have to necessarily balance brevity with presenting a complete picture of the state of the art in the multiple research areas that we draw upon. The comments of Reviewers 3 and 4 suggest that we may have been overly brief in some areas.

The result of these edits is a significant rewrite of the introduction to more clearly reflect the state of the art, in terms of both care and transmission, as well as clearer caveating of our discussion, owing to the assistance of the reviewers. We believe that this divides up our article more neatly between the surrounding context and our theoretical contributions.

(Comment 2.3) ...The results section is difficult to follow to a non-expert in mathematical model and again is very rich in speculative sentences that are not supported by any experimental data (e.g., metagenomic data about human gut microbiota).

Authors' reply: We appreciate that the reviewers collectively found the mathematical argument difficult to follow. Unfortunately, we are not certain that removing parts from the results would make the argument clearer to the non-expert, and we feel that moving the additional detail in the methods would make the main text substantially

harder to read for the non-expert. Where possible, we have tried to tweak the text to be more readable.

Additionally, while we have already acknowledged that our study is entirely theoretical and in silico, we are not entirely sure which sentences are speculative beyond interpreting our model results or referencing existing literature. Given our main method is the mathematical model, we also are hesitant to introduce a thorough discussion of metagenomic data beyond what we already mention for fear of making the paper significantly more review like (in the discussion: “metagenomics has shown that complex microbial communities are present in mammary milk...”).

Reviewer #3 (Remarks to the Author):

I liked this paper a lot, and I enjoyed reading your new and innovative ideas. One of the major strengths of the paper is also something that may pose some problems. The article essentially provides a new framework by which to better understand parental investment differences in infants, without having to rely on existing sexual selection hypothesis to justify why females invest more and males generally invest less. I like this, and I think we are overdue for some fresh perspectives on the unequal investment in offspring by different sexes...

(Comment 3.1) ... The authors do briefly mention Trivers parental investment theory, and mention that their work adds to this paradigm, so it is not in opposition to it. That's fine but give us the details on this and expand what you mean. I think the authors need to better situate their theory within the existing paradigm of parental investment and sexual selection theories, in both the introduction and in the discussion. This will make the main ideas put forward even more relevant and more clearly influential for the evolutionary biology community. Not making this more explicit may make it difficult for your theory to be immediately accepted alongside the existing frameworks. People can speculate about how your ideas fit in with major paradigms in evolutionary biology but might as well just do the work for them, and tell us exactly how framework fits within the existing paradigm of sexual selection driving differences between males and females in parental investment. I think doing this will help your paper have an even bigger impact on the biological community!

Authors' reply: This is a very good point. As addressed in our response to Reviewer 1 (Comment 1.3), it would be possible in future work to combine our modelling framework with those used more broadly for investigating the evolution of sex-biased parental care. For the time being we have attempted to tie our work more explicitly to modern theories of parental investment and sexual selection.

In the Introduction we now describe the key conditions that theoreticians have identified for the evolution of biparental care. We also state explicitly that we are interested in the subset of situations where biparental care has evolved, but male lactation has not. Meanwhile in the Discussion we now refer back to these points in the penultimate paragraph.

(Comment 3.2) Another issue in the paper is the consideration of prolactin, as it often can be very high in males of cooperative breeding species, or where paternal care is high. And yet males still do not lactate. So the consideration of prolactin levels to explain the lack of lactation needs to be fleshed out a bit better, given what we currently know. I provided several comments on this throughout the paper, so hopefully that will give you some ideas on how to better explain the mechanism behind the lack of lactation in males... I mentioned the possibility of the interaction between physiology and hormones, although I know this idea might be a anthropomorphism since we expect many females to grow mammary gland

tissue but not males. Not sure about differences in breast/chest physiology between males and females in other mammals.

Authors' reply: Thank you for raising this. We are not experts on prolactin, and are conscious of the need to be cautious in our statements on the subject. The correlation between prolactin levels and paternal care, is of special interest. We have made changes to the text as we describe below in the context of your points (see 3.6, 3.7, 3.12 and 3.16).

(Comment 3.3) Lastly, some of the language can be adjusted to be more inclusive. Natural birth could be changed to vaginal birth. Breast feeding could be changed to breast and chest feeding.

Authors' reply: Thank you for the suggestions. We are quite happy these got flagged in review and have tried to change these throughout.

See my other detailed comments below:

INTRODUCTION:

(Comment 3.4) 2nd sentence in intro: change "which" to "that"

Authors' reply: Done.

(Comment 3.5) Genetically male mammals... Could it be genetically and phenotypically male mammals? I am trying to think of situations where you can have genetic males that are phenotypically female (i.e. intersex).

Authors' reply: We appreciate that this is an important point and thank you for pointing it out, but we felt it is inappropriate to use the word "phenotypically" as the focus in reference 6 (in that sentence) is on trans women, while reference [5] notes lactation amongst cis men.

(Comment 3.6) Another thing that I think needs to be mentioned in the intro around the discussion of monogamy and paternity certainty, is that even in monogamous or polyandrous species, like in callitrichids, where males provide a lot of care, prolactin levels do increase a lot in these males, right when the female become pregnant, and they remain elevated for these males throughout the pregnancy and while the infants are small. The prolactin inhibits testosterone and helps induce paternal/caregiving behavior to the female and infant by the male. But if these males/fathers have higher prolactin, how come they are not lactating? Their prolactin levels are not high enough to induce lactation? So, the mechanism of higher prolactin is there too for the males in the monogamous species but it does not lead to lactation and nursing of young. Could be useful to add and consider because I think it would further support your point, that there seems to be a selection pressure against lactation by males, even in species that have all of the behavioral and hormonal factors set in place that would make male lactation possible. They still don't do it!

I just checked my PowerPoints because I teach this concept in one of my classes. The fathers in callitrichids actually have prolactin levels that are equally as high as the mothers. Also, more experienced fathers have higher prolactin levels than less experienced fathers. Sorry I can't provide you with the actual references because I can't remember them...

Authors' reply: We were unaware of this feature of callitrichids. We have now drawn the reader's attention to this in paragraph 9 of the Discussion, citing the paper by Almond et al. (2006).

(Comment 3.7) "restoration of normal prolactin levels". What is normal? Implying lactation is not normal... Change this sentence to be more neutral. Maybe "reversible once prolactin levels decrease again after lactation for the current offspring ends".

Authors' reply: We removed this sentence as we have added extra explanation about symbiosis to this paragraph.

(Comment 3.8) such elements is confusing. What elements. Be specific again: gut microbiome elements?

Authors' reply: We have changed both instances of "such elements" and added a referenced statement about the general identities of the elements we are thinking of in the fourth paragraph of the introduction.

(Comment 3.9) When you talk about vertical transmission of gut microbiome elements, I have some questions. What is vertical transmission? Define it. Why not just transmission? Is it because it is not horizontal transmission? So I guess explain vertical transmission enough to understand why you use that term. Also, vertical transmission of the gut microbiome... who's gut microbiome? From the parent/adult to the infant? Make clear.

Reading the rest of the paper, I now think you should certainly define in the intro your terms of vertical versus horizontal transmission.

Authors' reply: We appreciate these concerns (and those of Reviewers #4, see 4.2) about clarity and have tried to make our usage of terminology more precise. We now explicitly use referenced definitions of vertical and horizontal transmission as used in the symbiosis literature to the fourth paragraph of the Introduction.

(Comment 3.10) Same idea for the next several sentences where you talk about the host? You mean the host population of the infant/offspring, so the microbes that the infant is already developed in-utero?

Authors' reply: Similarly, we have tried to make clearer where host refers to a general individual, a parent, or an offspring, especially in the fourth paragraph, as well as trying to more clearly say we are focused on symbiont transmission through mammary milk, where there is a well-documented microbiome passing from the host mother to the host infant (Oikonomou et al. 2020). We also note that there are suggestions in the literature that some symbiont transmission could take place in-utero, but this is not yet well documented.

Also, what are symbionts? Maybe this term is regular for most people but some may read this and not quite know... is it different just from the term symbiote?

Authors' reply: We use 'symbiosis' as it was originally defined by de Bary (1887) and as used in the contemporary literature on symbiosis Douglas (1994, page 6), as a close association between two dissimilar organisms, these being a mammalian host and an element of the milk microbiome in the context of this paper. The association may be commensal, parasitic or mutualistic. A symbiont (= symbiote) is one of the partners in the symbiosis. In keeping with the literature on symbiosis (Douglas 1994, page 9), we use 'symbiont' as a neutral term to refer to the microbial element, and 'host' to refer to the mammal which carries it. We have added a sentence to the third paragraph of the Introduction to make clear what we mean by the term symbiosis.

(Comment 3.11) "switched off", "has remained switched off"- is a bit odd... Is there an alternative that doesn't sound as if we're talking about a light switch or turning a machine on and off? Maybe the word "suppressed" or "keep male lactation from happening", or "from being expressed", etc.

Authors' reply: We apologise for the insensitivity; we were speaking in terms of the model, but appreciate the connection between the model and what is being modelled. We have removed our usage of the word "switch" throughout and adopted "suppressed" instead.

(Comment 3.12) "do impair male activity"- could you be more specific as to what male activity you are referring to? Be specific here as to what male reproductive activity you are talking about? Also maybe say why? Prolactin inhibits testosterone which blah blah? Just an example, I do not know the mechanism.

Authors' reply: We removed this sentence as we have had to add extra explanation about symbiosis to this paragraph.

(Comment 3.13) infants born naturally, or natural birth needs to be changed. You mean infants born vaginally, or infants that exit through the birth canal.

Authors' reply: Agreed, thank you for the better phrasing.

RESULTS AND METHODS:

(Comment 3.14) I cannot comment on any of the algebraic models because I am far from a math expert (I am a biological anthropologist). The explanations of the models seem good/make sense from the results section explanations, although I admit I sometimes lost track of the reasoning, which is mostly because this level of math is difficult to assimilate for someone who doesn't use it often.

I hope another reviewer will be able to comment in detail on the logic of the algebraic models. Sorry, it's beyond my strengths

Authors' reply: We appreciate your attempt to engage with the mathematical models. To help, we have tried to modify the wording and mentioned how some of the

analysis was done in the Methods, but we are aware that adding too much explanation might make it harder to follow.

DISCUSSION:

(Comment 3.15) At times, it is unclear if you are talking about one of the parents or the infant/offspring. Can you try and make clearer throughout. For instance, in this sentence:

"We cannot prove that the danger of invasion by deleterious elements in the gut microbiome is the reason why there is no male lactation in mammals, so we put forward this argument as a hypothesis.

Can you add either "infant" or "offspring" before "gut microbiome".

Authors' reply: We are thinking of 'invasion' at the level of the host population, rather than at the level of colonisation by a microbe into a single host offspring. We have rewritten this sentence in paragraph 1 of the Discussion, to try to make this clearer. We have also been through the paper to check that, where 'invasion' is referred to, it is clear that this refers to the host population.

(Comment 3.16) Low prolactin level in males may be the case for most species, but again, doesn't seem to be the case for some cooperative breeding species like the callitrichids, or even humans, where fathers show high prolactin. Thus, in the discussion, I am not sure if genes for low prolactin levels in males is the best way to explain the absence of lactation in males.

Authors' reply: Thank you for raising this. We have now weakened our language in the fourth paragraph of the Discussion where we suggest a mechanism and have also now drawn attention to the paper by Almond et al. (2006) as a caveat to our proposed mechanism suggested by callitrichids in paragraph 9 of the Discussion.

(Comment 3.17) Careful that sometimes you use word "which" when it should be the word "that". Example, for the phrase: "operating through host modifier genes which control these levels" should be "that", not "which".

Authors' reply: We have now reviewed each usage of "which" and "that" in the text just in case.

(Comment 3.18) Unruly mob of microbes is delightful. I like it but is it too casual/ jargon-y? If nobody else has a problem with it, I am fine with it too.

Authors' reply: We agree, but, as Reviewer #3 expected, Reviewers #4 also found it too casual. We have removed the reference to "an unruly mob".

(Comment 3.19) If you wanted to be more inclusive, you could say breast and chest feeding when referring to breastfeeding in humans... Although some might argue that this is not necessary. I like to say it that way when possible, but up to you authors.

Authors' reply: Thank you for the suggestion. We have tried to make this change where appropriate.

(Comment 3.20) The fruit bat example is very interesting. I was also thinking that in humans, it is possible to make male lactate, isn't it, with domperidone to increase prolactin levels a lot? I would also say transgender females, but I guess they would have been exposed most likely artificially to progesterone and estrogens hormone replacement so that would affect chest/breast physiology ahead of lactation.... Hmm. But with Cis males, it is just that with a cis male physiology, one would need prolactin levels that are so much higher than for lactating females, just to induce milk synthesis in male glands? So maybe the idea of prolactin is more about the interaction between physiology and hormones? I don't know. Maybe that's too much to consider in this paper... Certainly your ideas about prolactin genes and co-evolution with the symbionts are interesting but details need to be ironed out a bit.

Reviewer: Iulia Badescu

Authors' reply: As addressed in our response to Comment 3.2, we have changed the paper to be more cautious in our discussion of specific hormonal regulatory systems. However we are pleased to see from this and the other reviewer comments that our article has triggered so many interesting ideas, and we agree that there are many interesting avenues to explore in future.

Reviewer #4 (Remarks to the Author):

We enjoyed reviewing this thought-provoking manuscript that brings together evolutionary theory, the microbiome, lactation, and mathematical modeling. We believe that the authors have put forth an intriguing hypothesis, but we have several concerns about the approaches and frameworks used in the paper. As detailed below, we are concerned that some of the models' assumptions are not well supported, and we believe that the manuscript lacks definitions of multiple key terms. We provide more detailed points below.

4.1. We are unsure whether the goal is to explain the absence of male lactation in humans specifically, or in mammals more generally. The human microbiome is different from other mammals and different still from the proto/early-mammals in which lactation first evolved, making this distinction important. Relatedly, the manuscript is also lacking in certain components of the broader evolutionary context.

Authors' reply: We understand how we caused a bit of confusion as to whether our focus was the class Mammalia or humans specifically. Our interest is in the Mammalia. Milk production is a defining feature of mammals, and is almost always absent from males. Nonetheless, we quite often refer to humans for two reasons. First, research on the human gut microbiome is more advanced than research on most other mammals. Second, we think the work on humans gives added interest to the paper for non-specialist readers.

We have tried to make this point clearer in the text by removing the reference to humans at the beginning of the abstract and making the first mention of humans in the text a reference to human milk.

4.2. The manuscript would benefit from clear definitions in the Introduction, as this would help the reader better understand the framework and approach. Definitions are needed for:

4.2.a. Symbiont – is any microbe in milk automatically considered a symbiont? How do the authors delineate between related ecological relationships like symbiosis, commensalism, and mutualism?

Authors' reply: Yes, any living element of the milk microbiome is taken to be a symbiont. As mentioned in our reply to Reviewer 3 (Comment 3.10), we have added a comment to the third paragraph of the Introduction that we use the term 'symbiosis' in its original sense, as it was defined by de Bary (1887), as a close association between two dissimilar organisms. This definition is in general use in the contemporary symbiosis literature (Douglas 1994 page 6). Importantly, the nature of the association is not defined: it may be mutualistic, commensal, parasitic, etc. The symbiosis literature typically refers to the microbes as 'symbionts', and the organism which houses the microbes as the 'host' (Douglas 1994 page 9). The effect of the symbiont on its host is defined by the relative fitness of the host when the symbiont is present. The numerical results on the symbiont sieve draw random values from a normal distribution for relative fitness, the distribution being centred on the point of

neutrality (=1), as shown in Fig 2a. For clarity and consistency with this literature, we have retained this nomenclature.

4.2.b. Vertical and horizontal transmission – it seems that the authors intend to use ‘vertical transmission’ to mean any microbial transmission from parent to offspring. We question this usage, since microbes that are transmitted from parents to offspring are not passed down through the same pathways as inherited genes are. Instead, one could argue that all maternal-offspring microbial transmission is horizontal and/or environmental— since transmission is thought to start at birth (and not in utero, as the authors note), then one could perceive microbial transmission to occur within the ‘birth environment,’ just as milk microbes could be shared through the ‘feeding environment.’

Authors’ reply: Again, we appreciate the concern about lack of precision and we have tried to make precise and justify our usage for Reviewers #3 and #4. We use the term ‘transmission’ rather than ‘inheritance’ to distinguish the passage of symbionts from parents to offspring from the much more controlled chromosomal path of genes from parents to offspring. Although ‘vertical transmission’ of symbionts is often through the host germ line, this is too restrictive because there are other ways in which a symbiont can move from host parent to offspring. An aposymbiotic phase in the host offspring is consistent with vertical transmission; in other words there can be a time lag following fertilization before host offspring acquires a symbiont from the host parent (see Bright and Bulgheresi Nature Reviews Microbiology 2010). The term ‘vertical’ is widely used in the transmission of the milk microbiome from host parent to offspring. See, for example, the review by Wang et al. 2020: “Maternal vertical transmission affecting early-life microbiota development”.

We have rewritten paragraph 4 of the Introduction, to provide further information, asked for by several reviewers. In doing this, we have added a comment on how we use the terms ‘vertical’ and ‘horizontal’ transmission.

Moreover, if the authors intend to use vertical transmission to mean parent-offspring transmission, then they should also be concerned with microbial transmission that occurs via skin-to-skin contact or the exchange of oral microbes. In other words, there are many routes of microbial transmission to offspring beyond milk, including behavioral routes that, like breastfeeding, have likely been shaped by natural selection (e.g. allocare/cooperative breeding; sociality and group living). If the authors want to focus on vertical transmission, it needs to be clearly defined and supported.

Authors’ reply: We apologise; we did not mean to imply that vertical transmission of symbionts to the infant gut occurs exclusively through the milk microbiome. Clearly that would be incorrect, and we have tried to be careful (e.g. in paragraph 4 of the Introduction) to now stress the period over which vertical transmission is taking place (from near to birth to weaning), that mammary milk is just one route for vertical transmission, and that it plays an important part in the infant’s gut microbiome. We have also cited the paper by Bright and Bulgheresi (2010) in paragraph 4.

Our study is concerned with the absence of milk production in male mammals, so we think it is reasonable that mammary milk should be the focus. There is still much to be learnt about the role of the milk microbiome, but the evidence is that milk

microbiome is a major contributor to the gut microbiome of the infant: 1000 to 10000 colony-forming units of bacteria per ml of healthy human milk is quoted in the literature (Jeurink et al. 2013).

4.2.c. Deleterious microbe – does this term apply to all types of microorganisms? The authors briefly mention viruses, but studies of the milk microbiome are often focused on bacteria. It is not clear which types of microbes, deleterious or otherwise, are most likely to be transmitted under the pathways and models presented here. We are concerned about the binary delineation between “good” and “bad” microbes, which often does not apply to complex microbial communities.

Authors' reply: We appreciate that the role of microbes is likely to be context-dependent in a microbiome, and it is the mapping from the effect of the whole microbiome to host fitness that matters. In this sense, we are using 'deleterious' as a shorthand to refer to symbionts that (cause the microbiome as a whole to) induce a decline in fitness in the host relative to the original state (i.e. $w < 1$ instead of $w = 1$ and that w is continuous rather than binarily good or bad).

As for which specific microorganisms might be of interest, as far as we know, a definitive answer cannot be given at the moment in the case of viruses. Still, there is no reason *a priori* why mutations with deleterious effects on their hosts should not occur in other taxonomic groups of the milk microbiome, such as fungi and viruses. However, if transmission of the microbiome is predominantly through maternal hosts (i.e. via the symbiont sieve), it is unlikely that deleterious microbes would be observed.

4.3. Relatedly, we also question why the authors chose to exclude host-to-host horizontal transmission. The social environment is a known pathway of microbial sharing across myriad host species (see recent studies on lemurs, howler monkeys, baboons, chimpanzees, and humans, as well as birds and bees), and in the context of this paper, we consider the social environment as a critical source of potential pathogen exposure for offspring. Further, it is not clear to us why the authors assume that milk is such a critical source of potential 'deleterious' microbial transmission, when it is well known that other routes of pathogen sharing, including those from conspecifics, pose considerable health risks to hosts. Given the evolutionary framing of this manuscript, the omission of well-known selection pressures that stem from socially transmitted pathogens is concerning.

Authors' reply: Reviewers #4's comment here is an interesting one, but one we approach from the opposite direction. 'Infectious' pathways are well-studied in mathematical modelling literature and, while relevant as Reviewers #4 say, did not help us clarify the story of uniparental lactation. As including two routes of horizontal transmission would seem to risk overcomplicating the story (whether including both in the results and discussion or by including both in a single model), we chose to only include environment-to-host transmission. With that having been said, we can produce figures equivalent to Figure 5 easily.

In the below (Mathematica generated) images (host-to-host transfer rate on the y axis, novel symbiont induced fitness on the x axis, biparental left, maternal right), the blue regions are parameter combinations that exclude the novel symbiont, red

regions contain only communities that include the novel symbiont, black regions have the population go to extinction, and orange regions are bimorphic. As Reviewers #4 might expect, without environment-to-host transmission, the novel symbiont can be completely excluded from the population (hence the new blue regions). There is also little to no inter-play with the symbiont sieve. In comparison to Figure 5 in the text, there is no (stable) bimorphic region in the biparental case, meaning that the system would only experience selection during transitory dynamics. We concluded from these images that the social environment is either too effective at spreading what can be spread (in which case the symbiont sieve is overwhelmed and we are dealing with epidemiology instead) or ineffective (in which case the symbiont sieve is unnecessary). Reviewers #4 are correct, social transmission is “critical”, but its inclusion complicates the model while not interacting with uniparental vs biparental transmission in a way that affects the key outcomes of the model.

Regarding milk as a “critical source of potential ‘deleterious’ microbial transmission”, we are perhaps viewing it from a slightly different perspective. We certainly agree that there are many other routes of pathogen sharing that pose considerable health risks to hosts (including host-to-host horizontal transmission). The selection pressures that these generate lead to a suite of evolutionary responses, from changes in mating systems (Kokko et al., 2002) to the maintenance of sex itself (Ashby and King, 2015). However our paper concerns the selection pressures that emerge as a result of vertical microbial transmission. We have shown here that vertical microbial transmission analogously generates a selection pressure against biparental inheritance, that in turn could explain the apparent absence of male lactation in mammals. Of course our model also predicts that under maternal transmission, milk is *not* a particularly critical source of ‘deleterious’ microbial transmission. The key point here is that if male lactation were to evolve, milk would have much greater potential as a route via which ‘deleterious’ symbionts could spread, and so male lactation is selected against under the mechanism of the symbiont sieve.

4.4. This paper makes a fundamental adaptationist assumption: that the absence of a feature is evidence of selection against the development of that feature. It must be

considered that the presence or absence of a feature is not necessarily adaptive, but may simply be unobtrusive enough to not be shaped by natural selection. Lactation and gestation are calorically expensive processes and must be worth the price to develop. Lactation first evolved in species where males had no role in the care or protection of infants, and it continued to evolve absent paternal care. The proposed selection mechanisms took place when the microbial composition of mammalian guts was very different than in humans today. To this day, paternal care—and particularly calorically expensive paternal care—is the exception, not the rule. Intensive parental care as seen in humans and in some primates is a recent evolutionary development, and reconfiguring a 250-300 million year-old system may not be feasible for a species like humans that has employed the cooperative care model for far less time.

Authors' reply: We appreciate the reference to classic literature (Gould and Lewontin, 1979 and the discussions of adaptationism that followed) and appreciate the need to provide evidence that the feature is in fact adaptive and not merely the result of neutral drift or another (truly) adaptive trait. We have a few things to note.

First, we do not claim in our manuscript that the absence of male lactation alone is evidence that it is selected against (which would be an adaptationist assumption). Instead, the section on "The symbiont sieve" establishes that male lactation has fitness effects and the section on "The evolution of the symbiont sieve" examines how these fitness effects can result in natural selection transitioning a population from biparental inheritance to maternal inheritance using host modifier genes. Figure 4 illustrates this as it takes place. The figure shows that a host population which carries a harmful symbiont generates selection for a host modifier gene that stops transmission of the symbiont from male hosts. In the example of Figure 4a, this selection is strong enough to drive the modifier gene to fixation (i.e. to complete removal of male lactation) before the harmful symbiont is able to get to fixation in the host population. In other words, restricting symbiont transmission to a single parent (females) is strongly advantageous in the presence of harmful symbionts. The underlying mechanism is an association which develops between host modifier gene and male hosts in which the harmful symbiont is absent (i.e. a positive coefficient of disequilibrium as shown in Figure 4b). This puts the host modifier gene that stops male lactation at a selective advantage over one which allows male lactation.

Second, we note that, as mentioned earlier (Reviewers #4, point 1), our paper is about the absence of male lactation in mammals in general, not just in humans. We do not require something uniquely human in order for our model to function. We note also that male mammals have retained the capacity to produce milk: milk production has been shown in Dayak fruit bats (Francis et al. 1994) and in humans (transgender women: Reisman and Goldstein 2018). So it would not seem necessary to reconfigure the entire system, which in turn would seem to be a lower hurdle.

Third, the above two points in combination with species that do have intensive parental care provides an adaptive reason as to why we should not see biparental lactation evolve despite the apparent opportunity. If the ability to lactate seems to be present in males that provide calorically expensive care, then we should see it begin

to appear unless there is a selection pressure against it. We believe our study identifies that pressure.

In summary, we claim that the absence of male lactation is a necessary consequence of evolution by natural selection, given the existence of: (1) harmful symbionts vertically transmitted in a host population, and (2) host modifier genes which can switch off host paternal transmission. We think this is a valuable discussion and have made reference to it at the beginning of our section on “The evolution of the symbiont sieve” in order to make our position more clearly “adaptive” and not “adaptationist”.

4.4.a. Additionally, in its own way, cooperative care supplements maternal caloric investment in the form of infant carrying and cosleeping. (See the work of Lee Gettler on male carrying and cosleeping)

Authors' Reply: This is a good point for future studies; we have now commented on this in the penultimate paragraph of the Discussion.

4.5. If the focus is to be on humans, this paper may be strengthened by framing it within the wider question of why humans (and some primates) do not seem to share the resource burden of lactation by employing allonursing as part of the cooperative care model.

Authors' Reply: As mentioned in our response to 4.1, our focus is not on humans specifically, but more generally on Mammalia. However we agree that this is an interesting point that lies parallel to the question of male lactation (see response to 4.5a, below).

4.5.a. In primates, evidence on allomaternal nursing and energetics is mixed. One study found that it is more likely to occur among cooperative breeders such as tufted capuchin monkeys (Hewlett and Winn 2014 - “Allomaternal Nursing in Humans” <https://doi.org/10.1086/675657>), but another found that it was more likely to occur when costs were lower—suggesting physiological constraints (MacLeod and Lukas 2014). Limited evidence suggests that although allomaternal nursing is widespread in humans, it is also rarely done—typically only in emergency situations, and typically only between kin. This begs the question: if it is rare, why?

Authors' Reply: In this paper we ask “*given that male parental care has evolved in many mammalian species, why has male lactation not followed suit?*”, and propose that selection pressures on microbial transmission as a possible answer. Similarly one could ask “*given that cooperative care has evolved in many mammalian species, why is allonursing far less common?*”. It is possible that a similar mechanism is at work, and our discussion of the allonursing literature was intended to point in this direction; indeed, as we describe, these systems may form an ideal testbed for the theory. However dealing with the issue of allonursing with the rigour that would be required lies outside the scope of this paper. First, the delineation between vertical and horizontal transmission becomes even thornier in this setting (see also, reply to 4.2.b). Second, a proper analysis of such systems would involve carefully accounting for the different types of possible population structures (e.g. group size, relatedness), social structures (e.g. intra-group variation in female reproductive rates) and

consequent care structures (e.g. “helper at the nest societies” vs “communal breeder societies”) (Konig 1997).

4.5.b. One observation that was made was that with the (limited) cross-cultural data was that allomaternal nursing was more common in cultures that lived in tropical climates, where infectious disease burden is higher, particularly parasitic infection (See Hewlett and Winn, 2014).

Authors' Reply: We agree that this is interesting, but, as the reviewers state, the data is limited and there might be other causes, such as maternal mortality. It would be interesting to see if this pattern played out across species, but this would need to be a future data-focused project.

4.5.c. Another observation (again, from limited data) was that the allonursers were slightly more likely to be paternal kin than maternal. This is interesting considering the increasingly demonstrated epigenetic and microchimeric effects of lactation.

Authors' Reply: This is another very interesting point that returns acutely to our response to 4.5.a; modelling microbial inheritance in the context of allonursing quickly gets exceedingly complicated, right down to the level of having to separate out the effects from maternal and paternal kin. We feel this is a programme of research in its own right to be addressed in the future.

4.6. In the discussion of male lactation in humans, this paper would benefit from a more in-depth discussion about alternative explanations for the absence of widespread lactation in males, and life history theory and cooperative care/biocultural care models. In humans, and to a degree in higher apes, lactation is intensely social and behavioral, and so biocultural contexts must be considered in explaining the presence or absence of a behavior, or of a developed mammary gland. Gestation and lactation are energetically expensive for the maternal body (particularly in humans), which would perhaps make male lactation beneficial regardless of the presence of deleterious microbial species, but at a cost to the male. In humans, that cost includes sharing the expensive caloric burden of an infant with a rapidly growing brain, and the homeostatic regulation provided by the skin-to-skin contact associated with breastfeeding. (See the work of Bogin, Bragg, and Kuzawa 2014 doi: 10.3109/03014460.2014.923938 and other recent work on cooperative care in the human lineage and non-human primates)

4.6.a. Additionally, lactational amenorrhea induced by breastfeeding (affecting only the female) leads to the wider interbirth intervals (compared to other primates) necessary to support the extended and calorically expensive early growth period characteristic of higher apes.

Authors' reply: We have now attempted to allude to this, and the other trade-offs that are ripe for evolutionary modelling, in the penultimate paragraph of the Discussion (see response to 4.4.a and reference to Gettler (2010)).

4.7. While we understand that all models are built on a set of assumptions, we are concerned about the assumptions used as the basis of this analysis. In particular, the first assumption (the fate of the new symbiont is independent of the resident microbiome)

violates fundamental principles of microbial community ecology (including the ecology of milk components and bacteria interacting within the infant gut). As supported both theoretically and empirically, microbial communities are inherently compositional, dependent networks. In fact, new bioinformatics and statistical approaches have been recently developed to address these core properties of microbial communities. As such, we question the validity and applicability of a model that is built on a violation of these well-known properties of microbial communities.

Authors' reply: We appreciate the reviewers' frustration that we have not considered the dynamics of interactions within the microbiome. As addressed in the section *Methods: Relation to other modelling frameworks*, accounting properly for ecological interactions in the microbiome together with vertical transmission and host-microbiome coevolution is a challenge for the modelling community in this area. This challenge is very much on our agenda, and probably needs to be dealt with through sequential assembly of microbiomes. But we realised that, in the space available, it would not be possible to cover this coherently, in addition to making the basic point about the symbiont sieve.

In view of this, and responding also to Reviewer 1, we have taken a step towards real-world complexities in revising the paper. We now show the symbiont sieve operating on a microbiome with 20 symbiont taxa (as opposed to a single taxon). We have taken a mean value of the 'w's of the symbiont taxa as the effect of the microbiome on host fitness. This is sufficient to show how important the symbiont sieve is for transmission of the microbiome as a whole. To make room for this we have replaced the original Figure 3 with some results on the multispecies microbiome, and replaced the original text on Figure 3 with new text. Our results suggest the sieve works with an arbitrary number of symbionts. We think the new results strengthen the argument for the importance of maternal transmission of microbiomes.

4.8. The paper is lacking in a discussion of the immune system, which has an intimate relationship to the microbiome. In fact, it has been posited that the genetic component of immunity may be a mechanism for the 'inheritance' of a microbiome. Lactation also confers immune benefits more generally. This is why, early in the COVID-19 pandemic, breastfeeding was recommended by the WHO even as very little was known about the virus and vertical transmission from mother to infant. A fundamental assumption for the authors' hypothesis is that the selective pressure exerted by the milk microbiome would have to surpass the other adaptive components of lactation, such as the profound immune benefits and the aforementioned defraying of caloric costs. We are not convinced that it does.

Authors' response: The reviewers raise an extremely interesting point with respect to the immune system and the immune benefits conferred by nursing.

We certainly agree that maternal milk has profound immune benefits, and can "prime" the neonatal immune system (with evidence for this extending to recognizing specific types of gut microbes (Rogier et al., 2014)). However in principle such benefits would also extend to paternal milk. This would actually provide another potential benefit to biparental lactation, with offspring being exposed to both maternal and paternal antibodies and thus an elevated adaptive immunity. For this reason we believe that a

consideration of the immune benefits associated with lactation strengthen our study, rather than weaken it.

However, we agree that it was remiss of us to not include reference to this important part of the lactation story in our original manuscript. In updating the paper, we have drawn attention to the importance of the immune system in milk in third to last paragraph of the Discussion, citing Rogier et al. (2014) which shows the continuing effects of SIgA, and the review by Chambers and Townsend (2020) on the protection given by human milk oligosaccharides.

4.9. A deeper discussion of lactation physiology would also be of benefit. The hormonal changes associated with pregnancy are essential to inducing the proliferation of lobules in the mammary gland that make milk synthesis possible. Although lactation without pregnancy is possible in humans, it is difficult to generate a full supply, and success is positively associated with gravidity. (See also: grandmother hypothesis and induced lactation/adoptive lactation) This suggests there have been evolutionary constraints operating on male mammary tissue development. It is also worth noting briefly in your paper, as not all readers may be aware, that males do in fact have some mammary tissue, and can develop breast cancer; their mammary tissue just doesn't develop in adolescence the way female mammary tissue does.

Authors' response: The physiology enabling lactation appears to be relatively straightforward to alter. For instance, Reisman and Goldstein (2018) describe the case of a trans woman who was able to achieve sufficient breast milk volume to be the sole source of nourishment for her child for 6 weeks. The infant was that of her partner, so milk production was not associated with gravidity. The success of such feeding suggests that the evolutionary constraints on development of male mammary tissues are not insurmountable.

We thank Reviewers #4 for making the point that some readers might not be aware of the presence of male mammary tissue. We have added this to the first paragraph of the introduction.

4.10. "Socially monogamous mammals" (Page 1, paragraph 2) – this is only about 3% of mammals. Lifelong pair bonding is relatively rare in mammals, and primates (including humans) are not more monogamous than other mammals. In highly social species, including humans, up to 20% of offspring are conceived as a result of copulations outside of the primary pair bond (See Reichard, U. H., and C. Boesch 2003. Monogamy: Mating strategies And partnerships In birds, Humans and Other Mammals. Cambridge University Press.)

Authors' reply: We have now rewritten the second paragraph of the introduction to make clear that we are particularly interested in the 10% of mammals in which biparental care of offspring has evolved, but male lactation as a caring strategy has not. We still include our reference to socially monogamous mammals in reference to the Azara's owl monkeys because we feel that this is a tangible example and that uncertainty in paternity is the most widely known factor for selection against paternal care arising from the parental investment theory literature.

4.11. “Gut microbiomes of infants born naturally” (Page 2, paragraph 3) – we strongly suggest using “vaginally” instead of “naturally.” “Natural” is unspecific and qualifying. It is mostly the vaginal microbes that are of interest here, but it is notable that mode of birth is not the only aspect of birth which may affect microbial transmission; others include the absence or presence of skin-to-skin practices, the surface and air microbial environment in hospitals, and the use of antibiotics.

Authors’ reply: Agreed, thank you and Reviewer #3 for pointing this out. We have opted for Reviewer #3’s phrasing.

4.12. Phylosymbiosis (page 6): there is conflicting evidence for the strength of this phenomenon across clades. We encourage the authors to review the literature and note any limitations.

Authors’ reply: Yes, we are aware that the match between microbiome composition and host phylogeny (phylosymbiosis) differs across the clades. Phylosymbiosis in the gut microbiome in general becomes less prevalent as microbiomes become taxonomically richer (Mallott and Amato 2021, Nature Review). The interesting point is that mammals are an exception: despite the complexity of their gut microbiomes, they commonly show evidence of phylosymbiosis. Mallott and Amato (2021) put this down to unique features of mammals, including mammary glands, viviparous birth, parental care of offspring and placentas. They point out that milk is a source of microorganisms delivered directly from the maternal body to the infant gut, allowing efficient vertical transmission. As a further step the authors note that bats are an exception to the pattern of phylosymbiosis seen in other groups of mammals. Our paper is consistent with these observations.

We have rewritten part of paragraph 6 of the Discussion to explain how our results are linked to phylosymbiosis. In short, our paper shows that maternal transmission of the milk microbiome acts as a symbiont sieve which helps to maintain the symbionts beneficial to the host. Slower turnover of the taxonomic composition of the microbiome is a likely outcome of this, and closer alignment with the host phylogeny is to be expected as a result.

4.13. The first sentence of the Discussion uses very strong, dramatic language. We suggest toning this down this sentence.

Authors’ reply: We have removed the reference to “an unruly mob”.

References

Ashby, B. and King, K.C., 2015. Diversity and the maintenance of sex by parasites. *Journal of evolutionary biology*, 28(3), pp.511-520.

Kokko, H., Ranta, E., Ruxton, G. and Lundberg, P., 2002. Sexually transmitted disease and the evolution of mating systems. *Evolution*, 56(6), pp.1091-1100.

Longo, G. and Soto, A.M., 2016. Why do we need theories?. *Progress in Biophysics and Molecular Biology*, 122(1), pp.4-10.

National Research Council, 2008. The role of theory in advancing 21st-century biology: catalyzing transformative research. *National Academies Press*.

Rogier, E.W., Frantz, A.L., Bruno, M.E., Wedlund, L., Cohen, D.A., Stromberg, A.J. and Kaetzel, C.S., 2014. Lessons from mother: long-term impact of antibodies in breast milk on the gut microbiota and intestinal immune system of breastfed offspring. *Gut microbes*, 5(5), pp.663-668.

Servedio, M.R., Brandvain, Y., Dhole, S., Fitzpatrick, C.L., Goldberg, E.E., Stern, C.A., Van Cleve, J. and Yeh, D.J., 2014. Not just a theory—the utility of mathematical models in evolutionary biology. *PLoS biology*, 12(12), p.e1002017.

Reviewers' Comments:

Reviewer #1:

Remarks to the Author:

The detailed and thoughtful response by the authors has addressed my concerns.

Reviewer #3:

Remarks to the Author:

I enjoyed reading this manuscript again, and I appreciate the work that the authors have put in to address the reviewers' comments and concerns. On my end, I am happy with the changes made and believe the authors have addressed all my questions and requested changes. Here are just two minor changes to be made to this improved version:

- Delete "breast and chest" in following sentence, as it is not necessary: "This matter is settled: metagenomics has shown that complex microbial communities are present in mammary milk [20], including human breast or chest milk [43, 44]."

It is enough to just say human milk.

- Replace "which" with "that" in following sentence: "It is notable that their microbiomes have more resemblance to those of birds which fly, showing little evidence of a correlation with host diet or phylogeny [50]."

Thanks and good luck,

Iulia Badescu

Reviewer #4:

Remarks to the Author:

We thank the authors for their line-by-line responses to our comments and questions. However, we continue to have reservations about certain aspects of this manuscript, which we discuss below.

OVERALL CONCERNS:

Overall, the paper lacks important context around evolutionary theory, community ecology, lactation physiology and the complexity of the lactation system and its influences. Some of these elements the authors deemed to be "beyond the scope" of this paper. However, this is an ambitious paper that is attempting to explain a core characteristic of the system that defines the class Mammalia. Therefore, the scope of the paper should reflect the scope and complexity of the phenomenon it is attempting to explain.

The hypothesis and model put forward in this paper depend on several assumptions which are not unequivocal and still have not been adequately justified by the authors:

- That the presence or absence of lactation (with adequate supply) is determined solely by genotype when we know it to be developmentally and environmentally influenced
- That the negative effects of the "deleterious microbe" would outweigh the benefits of biparental lactation, namely reducing the resource burden on the mother, greater food security and immunity.
- That male lactation was ever given the opportunity to be selected against in the first place
- That milk is the only path of transmission of potentially deleterious microbes
- That the presence of deleterious microbes would necessarily translate to selection on the system of

transmission and not just the parts of the system that determine the presence or prominence of the deleterious species

For these core reasons, we are unconvinced that the transmission of deleterious microbes would be the primary selective pressure against biparental lactation, and are concerned that this paper does not, on the whole, meet the publication standards for this journal. We hope our input above and specific suggestions below will help the authors improve their model and manuscript for publication in a more suitable venue.

SPECIFIC CONCERNS:

1. Authors did not address the broader evolutionary context including life history and development as requested in our comment 4.1. Lactation evolved in the gestating sex, and has existed absent direct paternal care for most of mammalian history. The question of why males don't lactate is in this way rather like asking why males don't gestate. In order for male lactation to be selected against, it must be introduced in the first place. The proposed deleterious microbes must have had the chance to be selected against, and as discussed in Item 3, direct paternal care would have been an opportunity for such microbes to be introduced socially via close physical contact. During the period when proto-lactation emerged, in egg-laying synapsids, direct paternal care, a presumed prerequisite for male lactation, was not likely the norm. The hypothesis tested here depends on something that likely never happened.

2. We asked the authors to define and explain how they are using the term 'symbiont' and their response is a citation for the original use of the term. As the authors note, the original citation is from 1887; we would prefer to see a contemporary reference for a term that is so integral to the manuscript (and appears in the title). In their response to our original comment about this point, the authors also referenced a 1994 citation that they refer to as the "contemporary symbiosis literature." Seeing as this citation is nearly 30 years old, we encourage the authors to update their definitions and citations. More specifically, we recommend contemporary microbiome literature that discusses the different types of microbe-host relationships.

3. Similarly, we also asked for clarification of the term 'deleterious microbe.' We are not sure that the authors fully understood our point with this term. The authors note that "there is no reason a priori why mutations with deleterious effects on their hosts should not occur in other taxonomic groups of the milk microbiome, such as fungi and viruses." However, we are not convinced that microbes that have "deleterious effects on their hosts" are the product of mutations; this perspective invokes a sort of phylosymbiosis in which microbes are meant to 'serve' their hosts, and so a microbe that harms the host is by default due to a mutation. We are not convinced that this is the case, nor does the literature definitively state that host-associated microbes are under selective pressure to benefit their host. This argument suffers from 'host-centric' bias in which the genetic material and functional capacity of microbes is evaluated in relation to its benefit to the host.

4. We thank the authors for their discussion of the spread of deleterious microbes, but upon reading their response, we would like to reiterate our concern about this part of their argument. The authors introduce the potential danger of biparental transmission at the end of the Introduction. Here, they state that a rare, deleterious (to the host) symbiont can have a two-fold chance of transmission to offspring if transmission routes (milk feeding) are bi-parental. Our chief concern with this logic is that it implies that the milk microbiome would contain rare deleterious microbes that could only be transmitted via milk. In the context of paternal investment in which this paper is built upon, opportunities for paternal-offspring microbial transmission would be high. Whether we call this vertical, horizontal, or social transmission, the important point is that offspring would have ample opportunities to receive deleterious microbes from their fathers, even in the absence of male lactation. Thus, we are not convinced that the risk of transmitting deleterious microbes to offspring is the main selection pressure against male lactation.

5. Further, it is quite plausible that selection has favored dominant milk bacteria that are known to play a critical role in establishing in the infant gut microbiome and engaging with the developing immune system (e.g. Bifidobacterium in human milk). A benefit of these dominant beneficial bacteria is aiding in the colonization resistance of potential pathogens (see <https://www.mdpi.com/2571-5135/9/2/7>; <https://www.sciencedirect.com/science/article/pii/S0022347621012774>; <https://www.sciencedirect.com/science/article/pii/S1097276520301490>). In this scenario, pathogens in milk and/or the surrounding environment are suppressed by the “good microbes” in milk. In other words, risks from environmental pathogens (of which there are many for any mammalian host) may be mitigated by the presence of “good microbes,” rather than the absence of “bad microbes,” in milk. Within this system, male lactation could still evolve, as the risk to offspring fitness from deleterious microbes would be reduced by processes of microbial ecology such as colonization resistance, competitive exclusion, predator-prey interactions, etc. In fact, this idea is in part supported by the authors updated Figure 3, in which the model of 20 taxa shows that all taxa increase in frequency in a similar manner. This supports the idea that the underlying microbial ecology of milk is of critical importance—deleterious taxa could rise in frequency if accompanied by the beneficial taxa that suppress or otherwise outcompete those taxa through ecological interactions.

6. For the reasons outlined in comments 3 and 4, we are concerned with the authors’ dismissal of the importance of ecological interactions within the milk microbiome. They note that “...accounting properly for ecological interactions in the microbiome together with vertical transmission and host-microbiome coevolution is a challenge for the modelling community in this area.” If accounting for these real-world, biologically relevant, ecological interactions is too great of a challenge, then perhaps this sort of modeling approach is not appropriate for the evolutionary question that this paper attempts to address.

7. The authors fail to address the lactation physiology they think is being selected against. Without a proposed mechanism, it seems that it could be gestation that is being selected against rather than lactation. Lactation evolved in the gestating sex, and it is during gestation that mammary glands mature and lobules proliferate. The authors argue in their response that lactation physiology is “relatively straightforward to alter”— but this is not at all the case most of the time, and in the case of assigned-male-at-birth humans is generally not possible without biomedical intervention taking place over the course of years coupled with dramatic and difficult behavioral changes lasting 4-9 months. Even in cis females, induced lactation without pregnancy does not usually result in a full supply. Involving a lactation specialist earlier in the writing process would have been of benefit here.

Reviewer #1 (Remarks to the Author):

The detailed and thoughtful response by the authors has addressed my concerns.

Reviewer #3 (Remarks to the Author):

I enjoyed reading this manuscript again, and I appreciate the work that the authors have put in to address the reviewers' comments and concerns. On my end, I am happy with the changes made and believe the authors have addressed all my questions and requested changes. Here are just two minor changes to be made to this improved version:

- Delete "breast and chest" in following sentence, as it is not necessary: "This matter is settled: metagenomics has shown that complex microbial communities are present in mammary milk [20], including human breast or chest milk [43, 44]."

It is enough to just say human milk.

Authors' Reply: Done.

- Replace "which" with "that" in following sentence: "It is notable that their microbiomes have more resemblance to those of birds which fly, showing little evidence of a correlation with host diet or phylogeny [50]."

Authors' Reply: Done.

Thanks and good luck,

Iulia Badescu

Reviewer #4 (Remarks to the Author):

We thank the authors for their line-by-line responses to our comments and questions. However, we continue to have reservations about certain aspects of this manuscript, which we discuss below.

OVERALL CONCERNS:

Overall, the paper lacks important context around evolutionary theory, community ecology, lactation physiology and the complexity of the lactation system and its influences. Some of these elements the authors deemed to be "beyond the scope" of this paper. However, this is an ambitious paper that is attempting to explain a core characteristic of the system that

defines the class Mammalia. Therefore, the scope of the paper should reflect the scope and complexity of the phenomenon it is attempting to explain.

Authors' Reply: This paper is not and has never been a review article aiming to interweave "evolutionary theory, community ecology, lactation physiology and the complexity of the lactation system and its influences." Despite that, Reviewer 2 criticised it on the grounds of being too like a review article. We see no way to be less like a review article and include more background information. The paper already spans across a wide range of research areas including evolutionary theory, symbiosis, microbiomes, lactation, and mathematical modelling.

The hypothesis and model put forward in this paper depend on several assumptions which are not unequivocal and still have not been adequately justified by the authors:

- That the presence or absence of lactation (with adequate supply) is determined solely by genotype when we know it to be developmentally and environmentally influenced

Authors' Reply: We note that this comment refers to the genotype of males, because this paper is about the absence or presence of lactation in males. If lactation **in males** was influenced by development and environment, you would expect to observe it in males from time to time. But this happens very rarely: In mammals, lactation is almost always absent in males. Genotype evidently overrides development and environment with respect to the distribution of lactation between males and females.

- That the negative effects of the "deleterious microbe" would outweigh the benefits of biparental lactation, namely reducing the resource burden on the mother, greater food security and immunity.

Authors' Reply: This is a fair criticism of the earlier version of the paper. We have now carried out further research on the potential nutritional benefits of biparental lactation. The extra theory is described in detail in the Supplementary Information, and summarised in a new subsection: 'Biparental nutrition and the symbiont sieve', starting on line 370. This shows that, under reasonable biological assumptions, a host population with uniparental (maternal) lactation cannot be invaded by a host mutant that allows a low level of male lactation. This is because the damage to the symbiont sieve is greater than the benefit from the extra nutrition.

The previous version of the paper had a paragraph just before the end of the Discussion, pointing out that we had not dealt with potential nutritional benefits of male lactation. As we now have some results on this, we have removed this and put in a new paragraph earlier in the Discussion (lines 503 - 523). This paragraph still has the caveat that the new research does not deal with all the potential costs and benefits, but that it does open up a framework for doing so in future research (lines 515-523).

- That male lactation was ever given the opportunity to be selected against in the first place

Authors' Reply: See point 1 below on: Gestation, lactation and the ancestral state of male mammals

- That milk is the only path of transmission of potentially deleterious microbes

Authors' Reply: See point 4 below on: Fathers have routes for transmission of deleterious microbes to the gut microbiomes of their offspring other than through milk.

- That the presence of deleterious microbes would necessarily translate to selection on the system of transmission and not just the parts of the system that determine the presence or prominence of the deleterious species

Authors' Reply: We are not suggesting, and do not believe, that the presence of deleterious microbes **only** affects the system of transmission. However, the fact that these microbes generate selection on the host's system of vertical transmission has not previously been recognised and is important for understanding the gut microbiomes of mammals. See point 6 below: On the importance of ecological interactions in the milk microbiome.

For these core reasons, we are unconvinced that the transmission of deleterious microbes would be the primary selective pressure against biparental lactation, and are concerned that this paper does not, on the whole, meet the publication standards for this journal. We hope our input above and specific suggestions below will help the authors improve their model and manuscript for publication in a more suitable venue.

Authors' Reply: There are a number of misunderstandings in the comments by Reviewers 4.

It is not our intention to suggest that the symbiont sieve generates the **primary** selective pressure against biparental lactation. In fact, we start the paper by pointing out the prevailing view that uncertainty over who the father is, will often limit the extent of paternal investment in offspring (see paragraph 2 of Introduction, lines 26 - 36). To leave no room for doubt, we have rewritten part of the first paragraph of the Discussion (lines 455 - 462).

That said, there are many instances of mammals in which paternal care is well developed, yet male lactation is absent. Despite the comments of the reviewers, the evidence is that male lactation is possible, so its absence when paternal care is otherwise well developed needs an explanation. This paper provides a novel answer which has important implications for development of the gut microbiome of mammals.

It becomes evident from the comments of the reviewers below, that they have misunderstood the evolutionary model at a rather basic level (see point 3 below on: What is a deleterious microbe?). Specifically, this paper is not about the evolution of microbial taxa within the gut microbiome (lines 123 - 124). It is about the evolution of modifier genes in the host that determine whether vertical transmission of the milk microbiome by the host is through one or both parents.

SPECIFIC CONCERNS:

1. Authors did not address the broader evolutionary context including life history and development as requested in our comment 4.1. Lactation evolved in the gestating sex, and has existed absent direct paternal care for most of mammalian history. The question of why males don't lactate is in this way rather like asking why males don't gestate. In order for male

lactation to be selected against, it must be introduced in the first place. The proposed deleterious microbes must have had the chance to be selected against, and as discussed in Item 3, direct paternal care would have been an opportunity for such microbes to be introduced socially via close physical contact. During the period when proto-lactation emerged, in egg-laying synapsids, direct paternal care, a presumed prerequisite for male lactation, was not likely the norm. The hypothesis tested here depends on something that likely never happened.

Authors' Reply: Gestation, lactation and the ancestral state of male mammals

Reviewers 4 would like us to have written a different, much more wide-ranging paper covering life history and development. Although this would be interesting, it would be more of a review, which was neither our objective nor something acceptable to Reviewer 2.

Transfer of microorganisms from the gestating parent to the neonate is unavoidable during the birth process. This asymmetry between parents naturally links lactation to the gestating sex when vertical transmission of symbionts is primarily uniparental. The paragraph on lines 151 - 165 points this out, together with some evidence on microbial transmission during birth, and possibly before birth. The coupling of lactation to gestation is important, and we emphasise this in the abstract of the paper.

The reviewers must surely understand that the changes needed for gestation to take place in male placental mammals are not possible because that would require fundamental alteration of the mammalian bauplan – male placental mammals do not have a uterus. Male lactation, on the other hand, does not require fundamental change. In fact, it is well known that male lactation (galactorrhea) can actually happen in special circumstances. We therefore reject the argument that “asking why males don't lactate ... is rather like asking why males don't gestate”.

The ancestral state of male mammals

The ancestral state of biparental or maternal lactation in the host is a matter of conjecture. Our new research allowing for nutritional benefits of biparental lactation shows: (1) that a host population with uniparental (maternal) lactation cannot be invaded by a host mutant that allows a low level of male lactation, and (2) that a host population with fully biparental lactation cannot be invaded by a host mutant that reduces male lactation by a small amount. So, if the ancestral state was maternal lactation, then the symbiont sieve would be a barrier to the evolution of biparental transmission, under conditions that favour other kinds of biparental care. If the ancestral state was biparental lactation, selection pressures other than those considered in this paper would be needed to move host populations to maternal lactation. Uncertainty about paternity, the traditional explanation for maternal lactation, is an obvious selection pressure.

We prefer not to speculate about an ancestral state of lactation. Our focus is on the selection pressures which continue to operate through time, due to the continuing presence of deleterious symbionts. In this context, we show that maternal lactation is protected from invasion by host mutations that would allow biparental lactation. However, we have now included a paragraph at the end of our new subsection: 'Biparental nutrition and the symbiont sieve', to cover the points above (lines 431 -

447). In addition, we have rewritten lines 495 - 502, and lines 552 - 553, which might have been interpreted as implying a particular ancestral state.

2. We asked the authors to define and explain how they are using the term 'symbiont' and their response is a citation for the original use of the term. As the authors note, the original citation is from 1887; we would prefer to see a contemporary reference for a term that is so integral to the manuscript (and appears in the title). In their response to our original comment about this point, the authors also referenced a 1994 citation that they refer to as the "contemporary symbiosis literature." Seeing as this citation is nearly 30 years old, we encourage the authors to update their definitions and citations. More specifically, we recommend contemporary microbiome literature that discusses the different types of microbe-host relationships.

Authors' Reply: Definitions of the term symbiosis

We have dealt with this issue in paragraph 3 of the Introduction (lines 49 - 68) in a new version of the paper for our appeal, as follows:

"We call the association 'symbiotic' because this term is widely used to describe intimate and prolonged physical associations between dissimilar organisms—here a mammalian host and a microbial symbiont—irrespective of where the association lies on the mutualism-parasitism continuum [21, 22]. We note that the term symbiotic is sometimes used as shorthand for mutualistic symbiotic interactions in the microbial literature [23, but see 24], but we need the more general usage here, because this paper is concerned with how variation in the interactions drives natural selection on the host's vertical transmission of microbes."

There is in fact a long history of debate over whether the term 'symbiosis' should be restricted to mutualisms. In the context of this paper we need a term that reflects the close, long-term nature of host-microbe associations in the gut. At the same time, the term **must not** imply a particular kind of interaction, such as mutualism, because the symbiont sieve operates on a continuum in which microbes have effects on hosts from deleterious to beneficial.

Our usage of the term symbiosis, which does not imply a particular kind of interaction, is standard – see for instance the current Wikipedia page: <https://en.wikipedia.org/wiki/Symbiosis> (most recent revision: 16 Nov 2023). Fuller discussions in the context of mutualism are in Bronstein (2015) and Douglas (2015) now cited in our paper.

In view of the confusion over the term 'symbiosis', we follow the long-standing advice of the American Society of Parasitologists, of defining exactly what we mean by symbiosis in paragraph 3 of the Introduction (lines 49 - 68).

3. Similarly, we also asked for clarification of the term 'deleterious microbe.' We are not sure that the authors fully understood our point with this term. The authors note that "there is no reason a priori why mutations with deleterious effects on their hosts should not occur in other taxonomic groups of the milk microbiome, such as fungi and viruses." However, we are not

convinced that microbes that have “deleterious effects on their hosts” are the product of mutations; this perspective invokes a sort of phyllosymbiosis in which microbes are meant to ‘serve’ their hosts, and so a microbe that harms the host is by default due to a mutation. We are not convinced that this is the case, nor does the literature definitively state that host-associated microbes are under selective pressure to benefit their host. This argument suffers from ‘host-centric’ bias in which the genetic material and functional capacity of microbes is evaluated in relation to its benefit to the host.

Author’s Reply: What is a deleterious microbe?

In the context of this paper, a deleterious microbe is a microbial taxon which, when added to a host microbiome, leads to a reduction in host fitness, as stated in lines 133-137. We have added a note in the Introduction (lines 99 - 102), to stress this point.

Importantly, we are not dealing with a mutation-selection process operating in the microbiome. This is explicit in lines 123 - 124. Evolution within microbial taxa is outside the scope of this paper (although of interest in its own right). We are drawing microbial taxa from a pre-existing pool of species, and these taxa have different fixed effects on host fitness (denoted by the variable w). These points are spelt out in the first paragraph of the Results (lines 114 - 130). At no point do we suggest that microbes are **meant** to serve their hosts.

It looks as though Reviewers 4 have misunderstood our argument at a basic level. In fact, it is the host population, not a population of microbes, that evolves. The host does have a mutation-selection process in place operating through genetic modifiers of vertical transmission (our focus is on selection, once the mutant modifier gene is present). The symbionts generate selection on these genetic modifiers of the hosts.

Figs 4a and 5c show the interplay of forces operating on host genes that modify vertical transmission. Evolution is taking place in the host population through an increase in frequency of the host gene M^- . Hosts carrying the gene M^- are restricted to maternal transmission of the microbiome. As M^- increases in frequency, the microbiome S , which contains a deleterious symbiont taxon, is eliminated from the host population in Fig 4a. If there is continuing input of the deleterious symbiont from the environment (Fig 5c), microbiome S remains at a low frequency in the host population. In this case, M^- has a continuing selective advantage over host modifier genes that would allow male lactation. In other words, maternal transmission is uninvadable by biparental transmission (lines 327 - 330).

The interesting outcome is that host evolution of uniparental transmission, driven in the first place by the symbionts, eventually feeds back to the microbiome and determines some important properties of it.

4. We thank the authors for their discussion of the spread of deleterious microbes, but upon reading their response, we would like to reiterate our concern about this part of their argument. The authors introduce the potential danger of biparental transmission at the end of the Introduction. Here, they state that a rare, deleterious (to the host) symbiont can have a two-fold chance of transmission to offspring if transmission routes (milk feeding) are bi-parental. Our chief concern with this logic is that it implies that the milk microbiome would

contain rare deleterious microbes that could only be transmitted via milk. In the context of paternal investment in which this paper is built upon, opportunities for paternal-offspring microbial transmission would be high. Whether we call this vertical, horizontal, or social transmission, the important point is that offspring would have ample opportunities to receive deleterious microbes from their fathers, even in the absence of male lactation. Thus, we are not convinced that the risk of transmitting deleterious microbes to offspring is the main selection pressure against male lactation.

Authors' Reply: Fathers have routes for transmission of deleterious microbes to the gut microbiomes of their offspring other than through milk.

We are well aware that milk is just one of a number of paths to the gut microbiome of the infant, and have pointed this out in lines 77-79. However, the importance of the path through mammary milk should not be underestimated: in healthy women, it is estimated to be of the order of 1,000,000 bacteria per day (Jeurink et al 2013, Wang et al. 2020; see lines 80 to 83).

The “main selection pressure against male lactation” is generally thought to be uncertainty about paternity; we emphasise this in the second paragraph of the Introduction (lines 26 - 36). However, this leaves open the question as to why lactation remains maternal in about 10% of mammalian taxa that display other kinds of biparental care. Our reason for writing this paper is to suggest a novel process – the symbiont sieve – which gives protection from deleterious symbionts, when lactation remains maternal.

Our view is that the symbiont sieve supplements the standard explanation for the absence of male lactation, and we are concerned not to overstate its significance. We have made an adjustment to the text to stress this point (lines 455 - 462).

5. Further, it is quite plausible that selection has favored dominant milk bacteria that are known to play a critical role in establishing in the infant gut microbiome and engaging with the developing immune system (e.g. Bifidobacterium in human milk). A benefit of these dominant beneficial bacteria is aiding in the colonization resistance of potential pathogens (see <https://www.mdpi.com/2571-5135/9/2/7>; <https://www.sciencedirect.com/science/article/pii/S0022347621012774>; <https://www.sciencedirect.com/science/article/pii/S1097276520301490>). In this scenario, pathogens in milk and/or the surrounding environment are suppressed by the “good microbes” in milk. In other words, risks from environmental pathogens (of which there are many for any mammalian host) may be mitigated by the presence of “good microbes,” rather than the absence of “bad microbes,” in milk. Within this system, male lactation could still evolve, as the risk to offspring fitness from deleterious microbes would be reduced by processes of microbial ecology such as colonization resistance, competitive exclusion, predator-prey interactions, etc. In fact, this idea is in part supported by the authors updated Figure 3, in which the model of 20 taxa shows that all taxa increase in frequency in a similar manner. This supports the idea that the underlying microbial ecology of milk is of critical importance—deleterious taxa could rise in frequency if accompanied by the beneficial taxa that suppress or otherwise outcompete those taxa through ecological interactions.

Authors' Reply: On selection for "good microbes".

The reviewers need to be clear about the mechanism for selecting good microbes. Given maternal transmission of the milk microbiome in the host, the symbiont sieve does select microbes with beneficial effects for the host, so good microbes do accumulate. We therefore agree with the reviewers, and in fact cite the case of *Bifidobacterium* (starting on line 533). We have added a further comment to stress that symbionts colonising the gut at an early stage can influence the subsequent assembly of the gut microbiome (lines 538 - 541).

However, on the fast timescale of microbial dynamics, once microorganisms are in the gut, selection on traits beneficial to the host needs direct intervention by the host, for instance by biologically active compounds such as immunoglobulins (line 603 - 606). We are not aware of a driving force within the microbiome dynamics *per se* to benefit the host.

The reviewer's argument that Fig. 3a supports biparental transmission has not taken into account the consequences of host genetic modifiers of vertical transmission. If some hosts carry the M- gene, preventing transmission from fathers, we expect this gene to have a selective advantage over M+ (which allows transmission from fathers). In other words, the biparental transmission is not evolutionarily stable – the host population would evolve towards maternal transmission.

6. For the reasons outlined in comments 3 and 4, we are concerned with the authors' dismissal of the importance of ecological interactions within the milk microbiome. They note that "...accounting properly for ecological interactions in the microbiome together with vertical transmission and host-microbiome coevolution is a challenge for the modelling community in this area." If accounting for these real-world, biologically relevant, ecological interactions is too great of a challenge, then perhaps this sort of modeling approach is not appropriate for the evolutionary question that this paper attempts to address.

Authors' Reply: On the importance of ecological interactions in the milk microbiome.

The reviewers' comments misrepresent our argument. We point out the importance of ecological interactions within the milk microbiome on line 124, and have a whole section discussing how to deal with them ("Methods: Relation to other modelling frameworks"). We have not dealt with ecological interactions among the microorganisms in this paper for the sake of clarity. The consequences of biparental and uniparental transmission of the gut microbiome are not well understood, and are especially clear to see in the simple case we have described.

As a matter of methodology in modelling, one usually tries to understand each individual component before integrating them, rather than creating a large black box a priori and trying to understand how all the complex components interact after the fact. This article is a study of an individual component as a part of an incremental approach to understanding the complexity surrounding ecological dynamics of the gut microbiome.

7. The authors fail to address the lactation physiology they think is being selected against. Without a proposed mechanism, it seems that it could be gestation that is being selected against rather than lactation. Lactation evolved in the gestating sex, and it is during gestation that mammary glands mature and lobules proliferate. The authors argue in their response that lactation physiology is “relatively straightforward to alter”— but this is not at all the case most of the time, and in the case of assigned-male-at-birth humans is generally not possible without biomedical intervention taking place over the course of years coupled with dramatic and difficult behavioral changes lasting 4-9 months. Even in cis females, induced lactation without pregnancy does not usually result in a full supply. Involving a lactation specialist earlier in the writing process would have been of benefit here.

Authors' Reply: Lactation physiology

The reviewers appear to argue that a physiological mechanism that would allow milk production in male mammals does not exist. This is incorrect. Milk production can (and very rarely does) happen in male mammals. It is well-documented as galactorrhea in humans, arising from hormonal imbalance. For example, this was reported in prisoners of war in World War II (Kunz & Hosken (2008) [https://www.cell.com/trends/ecology-evolution/fulltext/S0169-5347\(08\)00346-7](https://www.cell.com/trends/ecology-evolution/fulltext/S0169-5347(08)00346-7)). It is also documented that milk volume can be great enough for feeding an infant. This comes from a transgender woman with breasts of Tanner stage V, who produced sufficient milk volume to provide all the nourishment for her child for six weeks under appropriate medication (ref. [6] in the paper). We have added a further recent reference documenting milk production by a transgender woman (ref. [8] in the paper).

We are not experts in lactation physiology. But our reading of the subject area indicates that hormonal control of lactation could be adjusted to allow male lactation to take place. It is certainly “relatively straightforward to alter”, when compared with the introduction of a uterus into males for gestation, which the reviewers liken to male lactation (Point 1 above), and for which there is no evidence. The fact that male lactation does not generally happen in practice needs an explanation, given (a) that it could happen, and (b) that so much other parental care is carried out by males in certain mammalian taxa.

As noted under point 1, uniparental vertical transmission of symbionts naturally links lactation to the gestating sex, because some transfer of microorganisms from the gestating parent to the neonate during the birth process is inevitable (see lines 151 - 165).

Reviewers' Comments:

Reviewer #5:

Remarks to the Author:

In this paper, the authors ask why lactation has not evolved in mammalian males, even in species where paternal care is well developed. They propose the vertical transmission of symbionts from parent to offspring as a possible explanation, and study this hypothesis by developing a mathematical model where the ability for males to transmit microbes through milk is free to evolve. Specifically, they show that uniparental transmission acts as a symbiont sieve, preserving the host population from the propagation of deleterious symbionts. The authors show that their conclusions hold even when taking into account the nutritional aspect of milk. In my opinion, this paper addresses an important and valid question in an original and innovative way, is scientifically rigorous and very well written. I have no major concern and am unsure why a fifth reviewer was necessary to bring in... Nonetheless, I very much enjoyed spending time on this paper, and I thank and congratulate the authors for a very nice read!

Below are minor comments I would suggest to address:

1. I am far from being a lactation physiology specialist (and I thank the authors for everything I learned about it with this read!) but a basic idea that comes to my mind as an alternative hypothesis to theirs is that the hormonal changes that would be required to induce lactation in males would enter in competition with other factors that are specific to male hormonal regulation. In the words of the model, the activation of gene M from M- to M+ could be down-regulated by other male-specific genes. Although that is beside the scope of this paper, could you maybe comment on that?
2. L 206-212: this "non-mixing" aspect with maternal transmission is very interesting, but I am wondering whether it could be clarified a bit. Figure 3.b only shows the frequencies of each symbiont in the whole population, but not the distributions of symbionts within hosts. Do you simply mean that with most symbionts present in small frequencies it will rarely happen to find two different symbionts in the same host, or do you mean something more specific, related to the observed and unshown distribution of symbionts in hosts? I would suggest to either rephrase and clarify slightly if this is an indirect deduction from fig 3b, or find a simple way to visualize the distribution otherwise.
3. L304-306 and L 745: Regarding the system of differential equations that is presented as representative of the mean behavior of the stochastic process: what insures this is indeed the case? There can be important discrepancies as soon as there are non-linearities in the system, which is the case here.
4. L 444: I am not sure to understand the sense of the word "derived" in this sentence – might be because I am not a native speaker. Do you mean biparental care is not the majority in mammals? I suggest to rephrase.
5. L 645: While I could check all the rest of the math in the manuscript, going from line 2 to line 3 of equation (3) resisted me. I found this to be correct only if gamma was equal to 1, which is not true in every case, if the definition of gamma put in the manuscript is correct, and if I was correct to calculate the average fitness \bar{w} as $\bar{w} = p\bar{w} + q$. If there is no mistake in line 3, I suggest to at least add some indications, as I am certain I won't be the only confused reader.
6. S1: You are using again a gamma parameter, which meant something different in the main text. Consider changing one of them.
7. Supplementary section A, eq. 5-10 (and same comment for the next section): why do you consider here the possibility both to gain and to lose the additional symbiont horizontally, while in the main text you restricted yourself to only gain? (equations to be compared with L 750 in the main text)

8. Figures S1 and S2 : Please add sigma and gamma after "transmission probability" and "lactation benefits" respectively, by the big black arrows, for easier readability.

9. Legend of figures S2 and S3: what exactly do you mean by " 10^4 generations" in this continuous-in-time model?

REVIEWER COMMENTS

Reviewer #5 (Remarks to the Author):

In this paper, the authors ask why lactation has not evolved in mammalian males, even in species where paternal care is well developed. They propose the vertical transmission of symbionts from parent to offspring as a possible explanation, and study this hypothesis by developing a mathematical model where the ability for males to transmit microbes through milk is free to evolve. Specifically, they show that uniparental transmission acts as a symbiont sieve, preserving the host population from the propagation of deleterious symbionts. The authors show that their conclusions hold even when taking into account the nutritional aspect of milk. In my opinion, this paper addresses an important and valid question in an original and innovative way, is scientifically rigorous and very well written. I have no major concern and am unsure why a fifth reviewer was necessary to bring in... Nonetheless, I very much enjoyed spending time on this paper, and I thank and congratulate the authors for a very nice read!

Below are minor comments I would suggest to address:

1. I am far from being a lactation physiology specialist (and I thank the authors for everything I learned about it with this read!) but a basic idea that comes to my mind as an alternative hypothesis to theirs is that the hormonal changes that would be required to induce lactation in males would enter in competition with other factors that are specific to male hormonal regulation. In the words of the model, the activation of gene M from M- to M+ could be down-regulated by other male-specific genes. Although that is beside the scope of this paper, could you maybe comment on that?

We thank the reviewer for raising this very important point. Our model of the genetic modifier is intended to be schematic, because in general there are many ways in which transmission of symbionts from males to their offspring could be limited. In the case of the hormonal control by prolactin in mammals, epistasis through interactions between genes and pleiotropy are indeed discussed in the literature as possible processes constraining its evolution at a molecular level. Interestingly, it is thought that the detailed molecular architecture allows continuing evolution in spite of the constraints caused by epistasis and pleiotropy (doi:10.1016/j.ygcn.2021.113791). Although we have not taken our model of evolution down to this detailed level, we think it important to draw the attention of readers to the molecular work, and have now expanded the text around L613-621 of the Discussion to make this clear.

2. L 206-212: this “non-mixing” aspect with maternal transmission is very interesting, but I am wondering whether it could be clarified a bit. Figure 3.b only shows the frequencies of each symbiont in the whole population, but not the distributions of symbionts within hosts. Do you simply mean that with most symbionts present in small frequencies it will rarely happen to find two different symbionts in the same host, or do you mean something more specific, related to the observed and unshown distribution of symbionts in hosts? I would suggest to either rephrase and clarify slightly if this is an indirect deduction from fig 3b, or find a simple way to visualize the distribution otherwise.

We thank the reviewer for raising this matter. We know precisely what the microbiome of each host individual is throughout the simulation. So the frequency distribution for the number of symbionts per host can be computed at any time. We have therefore put some extra information into Fig. 3 to show what is happening, and have added a small amount of text to explain this. Briefly:

- Start time (panels b, f): initial conditions are identical for the two modes of transmission, so the frequency distributions are the same.
- Intermediate time (panels c, g): frequency distributions are diverging.
- End time (panels d, h): close to the eventual (asymptotic) state.
- Asymptotic state (not shown): Biparental transmission has all symbiont taxa present in every host. Maternal transmission has one symbiont taxon present in every host, this being the taxon which confers the greatest benefit on the host.

3. L304-306 and L 745: Regarding the system of differential equations that is presented as representative of the mean behavior of the stochastic process: what insures this is indeed the case? There can be important discrepancies as soon as there are non-linearities in the system, which is the case here.

To make sure that our stochastic simulations matched the system of ordinary differential equations (ODEs), we constructed the ODEs from the corresponding master equation. This involved rescaling the master equation from the individual scale to densities using the scaling parameter ν and retaining the first order terms. To confirm that this was sufficient to capture the behaviour of the stochastic process, we checked for consistency between the stochastic realisations and the ODEs at various stages of our analysis.

To reassure the reader that our method has worked satisfactorily, we have added one such solution to the paper where we move from stochastic simulations to ODEs (Fig 5c). This figure is for the more complex case involving the host modifier gene for vertical transmission, which means we must also take explicit account of the abundances of males with and without the modifier gene.. As such, we have also written down the full, explicit system of six ODEs in the Methods section (Eqs 7 to 12).

4. L 444: I am not sure to understand the sense of the word “derived” in this sentence – might be because I am not a native speaker. Do you mean biparental care is not the majority in mammals? I suggest to rephrase.

“Derived” has a specific meaning in phylogenetics, referring to a character or feature found within a single lineage of a larger group which is not shared with all organisms in the larger group. For clarity, we have rewritten the sentence to avoid use of the term.

5. L 645: While I could check all the rest of the math in the manuscript, going from line 2 to line 3 of equation (3) resisted me. I found this to be correct only if gamma was equal to 1, which is not true in every case, if the definition of gamma put in the manuscript is correct, and if I was correct to calculate the average fitness \bar{w} as $p\bar{w}+q$. If there is no mistake in line 3, I suggest to at least add some indications, as I am certain I won't be the only confused reader.

Thank you for checking our mathematics and indicating this point of difficulty. We have now included the equation for mean fitness \bar{w} in the manuscript and have added some intermediate steps to make the line of reasoning easier to follow. These can be found on lines 657 to 662.

6. S1: You are using again a gamma parameter, which meant something different in the main text. Consider changing one of them.

Thank you for catching this, we have now replaced the gamma in the main text with phi.

7. Supplementary section A, eq. 5-10 (and same comment for the next section): why do you consider here the possibility both to gain and to lose the additional symbiont horizontally, while in the main text you restricted yourself to only gain? (equations to be compared with L 750 in the main text)

We were originally attempting to explain our reasoning on lines 48-53 of the Supplement, but we agree in hindsight that it was not entirely clear. We have now rewritten the paragraph in response (see SI lines 48-57).

The essential point is that when hosts can only gain the symbiont, there exist parameter regions where the symbiont can fixate. Comparing where these regions occur under biparental vs uniparental transmission is a useful exercise for understanding the benefits of maternal transmission (see Main Text Fig 5). However, because we only consider two symbiont communities in our mathematical model, these regions of fixation are equivalent to the microbiomes of all hosts being indistinguishable. This is a problem when we start considering evolution, as selection on the modifier genes that suppress male lactation can only operate if there is some variation in microbiome composition. In reality, this variation is supplied by the plethora of symbionts that can enter (and further evolve within) hosts. As our simple mathematical model has only two community classes, we must instead generate this variation by allowing both gain and loss of the symbionts.

8. Figures S1 and S2 : Please add sigma and gamma after “transmission probability” and “lactation benefits” respectively, by the big black arrows, for easier readability.

We agree that this is clearer and have amended the figures accordingly.

9. Legend of figures S2 and S3: what exactly do you mean by “ 10^4 generations” in this continuous-in-time model?

We thank the reviewer for pointing this error out, which emerged from a force of habit. We have amended the text to instead read “timesteps”.

Reviewers' Comments:

Reviewer #5:

Remarks to the Author:

I thank the authors for their detailed answers to my minor questions and comments. My comments previously numbered 1, 2, 4, 5, 6, 7 and 8 have been fully addressed. Below I would like to comment again on points 3 and 9, but I do not think these last comments necessitate any specific answer, as I think all scientific matter relevant for the scope of this paper have been addressed.

- Previous point 3: I realize my question was not very well formulated. As you rightly point out in your answer, in a system with non-linearities the system of equations for the moments is typically not closed, with the equation for the average size of the population depending on higher order moments that are not necessarily negligible. With your answer I understand what I could have understood right away but had previously overlooked, which is that rescaling and adopting a density description instead of following the actual population size allows you to get rid of these higher order terms. Thus, I apologize for this unnecessary question, but would like to point out that the match between one single stochastic realization and the ODE describing the average is not a good argument (I personally would keep the solution lines in Fig. 5 for visual aid but would get rid of the sentence line 766-768 - but that is just a personal preference).

- Previous point 9: If I would like to reproduce those plots, I still miss the size of the timestep... as I understand it, you solve up to $t=10^4$ and use d_0 set to 1 to scale time.

Additional comments :

- Fig. 3: I suggest to choose between "number [of] microbial taxa per host" and "number of symbiont taxa per host" for the labeling of panels b, c, d and f, g, h

- A thought that is completely beyond the scope of this paper, but just some curiosity I had for myself: although transmission of symbionts from parents to offspring is not completely akin to genetic inheritance, I was wondering if you had thought about how your model relates to models of other "non-Mendelian" inheritance processes like gene drive.

REVIEWERS' COMMENTS

Reviewer #5 (Remarks to the Author):

I thank the authors for their detailed answers to my minor questions and comments. My comments previously numbered 1, 2, 4, 5, 6, 7 and 8 have been fully addressed. Below I would like to comment again on points 3 and 9, but I do not think these last comments necessitate any specific answer, as I think all scientific matter relevant for the scope of this paper have been addressed.

- Previous point 3: I realize my question was not very well formulated. As you rightly point out in your answer, in a system with non-linearities the system of equations for the moments is typically not closed, with the equation for the average size of the population depending on higher order moments that are not necessarily negligible. With your answer I understand what I could have understood right away but had previously overlooked, which is that rescaling and adopting a density description instead of following the actual population size allows you to get rid of these higher order terms. Thus, I apologize for this unnecessary question, but would like to point out that the match between one single stochastic realization and the ODE describing the average is not a good argument (I personally would keep the solution lines in Fig. 5 for visual aid but would get rid of the sentence line 766-768 - but that is just a personal preference).

The reviewer is of course correct: it is the average of the stochastic realisations that should converge to the ODE solution. In Fig. 5 we want to make a visual link between the results of the stochastic model (Fig. 5c) and the results of the deterministic model that map the regions of parameter space (Fig. 5a, b). We have therefore followed the advice of the reviewer, taking out the sentence in lines 766 to 768.

- Previous point 9: If I would like to reproduce those plots, I still miss the size of the timestep... as I understand it, you solve up to $t=10^4$ and use $d0$ set to 1 to scale time.

Apologies, we had misunderstood the question. Equations 5-19 in the supplement are solved numerically between $t=0$ and $t=10^4$ to obtain Supplementary Figures 2 and 3. In terms of the size of timesteps, these are variable; we use Mathematica's NDSolve function which adapts its step size to achieve a specified precision in an efficient manner. We have now changed "for 10^4 timesteps" to "between $t=0$ and $t=10^4$ " in these figure captions, and added "Simulations are conducted using NDSolve in Mathematica (see Main Text Code Availability Statement)" to the end of the captions.

Additional comments :

- Fig. 3: I suggest to choose between "number [of] microbial taxa per host" and "number of symbiont taxa per host" for the labeling of panels b, c, d and f, g, h

We have removed the typo in the course of reconstructing Fig. 3 to span two columns. Both labels now read: "number of symbiont taxa per host".

- A thought that is completely beyond the scope of this paper, but just some curiosity I had for myself: although transmission of symbionts from parents to offspring is not completely

akin to genetic inheritance, I was wondering if you had thought about how your model relates to models of other “non-Mendelian” inheritance processes like gene drive.

Yes, there are interesting parallels to be made between the mechanisms of inheritance of genes and the mechanisms of vertical transmission of host symbionts.

Gene drive: Under biparental transmission of symbionts it is enough for one parent to carry a symbiont taxon for this to be present in all offspring. This is analogous to gene drive in which the 0.5 probability of Mendelian inheritance of an allele is replaced with a probability of 1. Uniparental transmission prevents this.

Recombination of genes: Biparental transmission allows free mixing of the symbionts from both parents. Uniparental transmission prevents any mixing from happening.

The mechanisms available for host control over vertical transmission of symbionts are much more restricted than those available for inheritance of genes. But uniparental transmission is enough to have some important effects.